# Predictive reward-prediction errors of climbing fiber inputs integrate modular reinforcement learning with supervised learning

**Huu Hoang**[1]ↂ*, **Shinichiro Tsutsumi**[2]ↂ, **Masanori Matsuzaki**[3], **Masanobu Kano**[4,5], **Keisuke Toyama**[1], **Kazuo Kitamura**[6]*, **Mitsuo Kawato**[7]*

**1** Neural Information Analysis Laboratories, Advanced Telecommunications Research Institute International, Kyoto, Japan, **2** Laboratory for Multi-scale Biological Psychiatry, RIKEN Center for Brain Science, Saitama, Japan, **3** Department of Physiology, The University of Tokyo, Tokyo, Japan, **4** Department of Neurophysiology, The University of Tokyo, Tokyo, Japan, **5** International Research Center for Neurointelligence (WPI-IRCN), The University of Tokyo, Tokyo, Japan, **6** Department of Neurophysiology, University of Yamanashi, Yamanashi, Japan, **7** Computational Neuroscience Laboratories, Advanced Telecommunications Research Institute International, Kyoto, Japan

ↂ These authors contributed equally to this work.
* hoang@atr.jp (HH); kitamurak@yamanashi.ac.jp (KK); kawato@atr.jp (MK)

## Abstract

Although the cerebellum is typically associated with supervised learning algorithms, it also exhibits extensive involvement in reward processing. In this study, we investigated the cerebellum's role in executing reinforcement learning algorithms, with a particular emphasis on essential reward-prediction errors. We employed the Q-learning model to accurately reproduce the licking responses of mice in a Go/No-go auditory-discrimination task. This method enabled the calculation of reinforcement learning variables, such as reward, predicted reward, and reward-prediction errors in each learning trial. Through tensor component analysis of two-photon Ca$^{2+}$ imaging data from more than 6,000 Purkinje cells, we found that climbing fiber inputs of the two distinct components, which were specifically activated during Go and No-go cues in the learning process, showed an inverse relationship with predictive reward-prediction errors. Assuming bidirectional parallel-fiber Purkinje-cell synaptic plasticity, we constructed a cerebellar neural-network model with 5,000 spiking neurons of granule cells, Purkinje cells, cerebellar nuclei neurons, and inferior olive neurons. The network model qualitatively reproduced distinct changes in licking behaviors, climbing-fiber firing rates, and their synchronization during discrimination learning separately for Go/No-go conditions. We found that Purkinje cells in the two components could develop specific motor commands for their respective auditory cues, guided by the predictive reward-prediction errors from their climbing fiber inputs. These results indicate a possible role of context-specific actors in modular reinforcement learning, integrating with cerebellar supervised learning capabilities.

**Data availability statement:** The customized MATLAB code for the analyses and the simulation using the CARLsim framework are publicly available on GitHub at the following link: https://github.com/hoang-atr/go_nogo.

**Funding:** HH, KK and KT were supported by Grants-in-Aid for Scientific Research in Innovative Areas (17H06313). MM, M Kawato and KK were supported by Grants-in-Aid for Transformative Research Areas (22H05160 to MM, 22H05156 to M Kawato, and 22H05161 to KK). HH and KT were partially supported by JST ERATO (JPMJER1801, "Brain-AI hybrid"). HH, M Kawato, and KK were partially supported by the Grant Number JP21dm0307002, JP21dm0307008, and JP19dm0207080, respectively, Japan Agency for Medical Research and Development (AMED). M Kawato was partially supported by Innovative Science and Technology Initiative for Security Grant Number JP004596, Acquisition, Technology & Logistics Agency (ATLA), Japan. M Kano and KK were partially supported by Grants-in-Aid for Scientific Research (JP18H04012, JP20H05915, JP21H04785 to M.Kano and JP22H00460 to KK) from the Japan Society for the Promotion of Science (JSPS). The funders had no role in study design, data collection and analysis, decision to publish, or preparation of the manuscript.

**Competing interests:** The authors have declared that no competing interests exist.

## Author summary

The cerebellum, best known for its role in sensorimotor functions and supervised learning, also contributes to learning from rewards. We studied mice performing an auditory discrimination Go/No-go task and used a mathematical model to calculate essential reward signals, such as reward-prediction errors. By analyzing two-photon calcium imaging data of climbing fiber inputs—key inputs to the cerebellum—we found they were closely linked to reward signals and helped the mice improve their licking responses under different contexts. To explore this further, we created a neural network model of the cerebellum, demonstrating how specific cerebellar neurons use reward-prediction errors to generate appropriate actions depending on the contexts. These findings reveal that the cerebellum plays a vital role in the modular learning and adaptive behavior based on rewards.

## Introduction

Historically, the cerebellum has been considered to implement supervised learning algorithms, as suggested by seminal works [1–13]. Recent research, however, has identified a potential role of the cerebellum in reward processing and reinforcement learning tasks driven by rewards and penalties [14–19]. Particularly, climbing fibers (CFs), one of the two primary input sources to the cerebellum, are associated with both sensorimotor variables and a broad range of reward contingencies [20–24], distinct from movement-related information [21,22]. Intriguingly, the association between cerebellar activities and reward variables is dependent on the cerebellar cortex's zonal organization [24,25].

In a previous study [26], we analyzed two-photon Ca²⁺ imaging data of climbing fibers from eight aldolase-C zones in the left Crus II of the mouse cerebellum [27,28] during a Go/No-go auditory discrimination licking task. Using tensor component analysis [29], we identified four distinct components that dynamically organized through different mechanisms to facilitate learning in precise timing control, error reduction, reward processing, and lick suppression. Given the reward-based nature of this task, our findings suggest that climbing fiber inputs may encode reward-prediction errors, providing critical feedback that modulates the strength and timing of Purkinje cell activity. This modulation may refine motor responses and decision-making processes. However, the exact mechanisms by which climbing fiber inputs guide reward-based learning remain to be fully elucidated.

Building on this prior work, the current study explores how these functional components interact to support reinforcement learning. Using the same data sets, this study introduces three key innovations to the previous analyses. First, we employed a reinforcement learning algorithm (Q-learning [30]) to model licking responses and compute key reinforcement learning variables, including rewards, predicted rewards, and reward-prediction errors. Second, we conducted systematic regression analyses to examine the relationship between climbing fiber activity across the eight aldolase-C zones and the four functional components, and reward- and sensorimotor-control variables on a trial-by-trial basis. To achieve this, we extended the tensor component analysis in the previous study to quantify the four functional components for each learning trial, which is crucial to elucidate the specific roles of climbing fiber inputs in behavioral learning. Our trial-based analyses revealed that climbing fiber inputs to the first two functional components, which exhibited bidirectional modulation during learning, were strongly and negatively correlated with predictive reward-prediction errors. As the third innovation, we developed a cerebellar neural network model comprising 5,000 granule cells, Purkinje cells, cerebellar nuclei

neurons, and inferior olive neurons, based on the assumption of bidirectional synaptic plasticity at parallel fiber-Purkinje cell synapses [31–35]. The model demonstrated that the two functional modules of Purkinje cells, roughly corresponding to aldolase-C zones, could generate distinct, context-specific motor commands based on predictive reward-prediction errors relayed through climbing fiber inputs. The significance of this framework lies in its potential to advance the development of new reinforcement-learning algorithms, enabling them to efficiently learn complex tasks within a limited number of trials [18].

## Results

### Q-learning model of licking behavior

We employed a Q-learning algorithm to model licking behavior of mice performing the Go/No-go task (Fig 1A). The Q-learning model was selected because it is one of the simplest

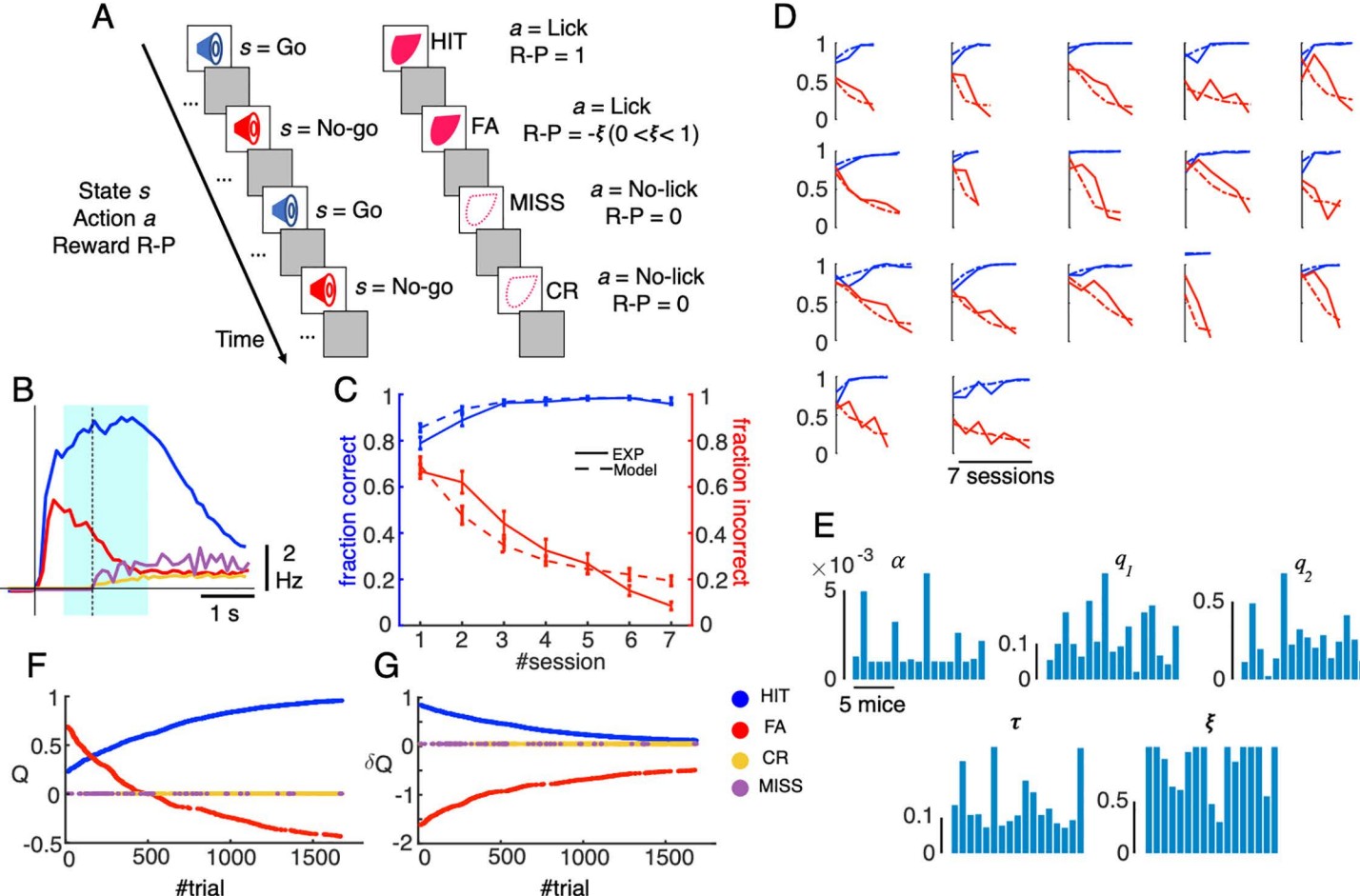

**Fig 1. Q-learning model of a licking behavior during Go/No-go auditory discrimination task.** A: schematic of the Q-learning model. B: the averaged lick rate for the four cue-response conditions (blue, red, orange and magenta for HIT, FA, CR and MISS trials, respectively: see also inset for color codes). Solid and dashed vertical lines indicate the cue onset and response window (1s after cue), respectively. Light cyan shading represents the window for possible reward delivery (0.5 - 2 s after cue). C-D: fraction correct for Go cue (blue) and fraction incorrect for No-go cue (red) for experimental data (solid lines) and Q-learning model (dashed lines), averaged for 17 mice in 7 training sessions (C) and for individual mice (D). Vertical bars in C show standard errors. E: hyperparameter values of the Q-learning model estimated for individual mice (1~17); and from left to right and top to bottom, learning rate $\alpha$, initial Q values for Go and No-go cues, $q_1$ and $q_2$, respectively, temperature $\tau$, and punishment value for FA trials $\xi$. F-G: evolution of Q (F) and $\delta Q$ (G) of a representative mouse for the four state-action combinations (HIT, FA, CR, MISS) during the course of learning.

reinforcement learning algorithms, based on state-action value functions rather than state value functions, and has a small number of hyper-parameters. Briefly, mice were trained to distinguish two auditory cues (high versus low frequency tones) by either licking or not licking during a response period of 1s after cue onset to obtain a liquid reward. The reinforcement-learning-algorithm state $s$ is determined by the two auditory cues ($s$ = Go/No-go), and the reinforcement-learning-algorithm action $a$ is determined by licking during a time window of $[0, 1]$ s after the auditory cue ($a$ = Lick/ No-lick). A reward value R-P was assigned according to cue-response conditions, including HIT trials (lick after Go cue) in Go task (R-P = 1), false alarm trials (FA, lick after No-go cue, R-P = $-\xi$ , $0 \le \xi \le 1$), correct rejection trials (CR, no lick after No-go cue, R-P = 0) and MISS trials (no lick after Go cue, R-P = 0). Note that the R-P value for FA trials was constrained to be negative because a lick after a No-go cue was punished with a timeout of 4.5s, while that of CR and MISS trials were 0 because neither reward nor penalty was given in those trials. The Q-learning algorithm assumed that a reward prediction Q was computed as a function of the two variables $s$ and $a$; *state and action*. In each trial, a reward prediction error $\delta Q$, a difference between reward R-P and Q, was used to update computation of Q in the next trial with a learning rate $\alpha$. For an action policy, we used the softmax function with a single-temperature parameter $\tau$ to convert Q values into a probability distribution over the two possible actions; Lick and No-lick. Since mice had undergone pre-training sessions lasting 3 days, when they were always rewarded for a lick within 1s for both cues, initial Q values for both Go and No-go cues were positive ($0 \le q_1 \le 1, 0 \le q_2 \le 1$, for Go and No-go cues respectively, see Methods for details). It is important to note that we did not explicitly model the time course within trial or temporal difference errors as in previous studies (e.g., [36]). Instead, we assumed a single timing of cue representation for computation of Q and $\delta Q$, which precedes the timing of reward delivery by about 0.5–2 s. Thus, the reward-prediction error $\delta Q$ in this study is a predictive signal estimated before the actual delivery of liquid rewards. We will later discuss implications of this assumption on timing for possible neural computations. Briefly, we show that the cerebellar reinforcement learning in this study could acquire the actor but not the critic through temporal difference error.

Behavioral data indicated that the lick rate in HIT trials was high in the early period (average and standard deviation of lick rate and lick latency, 4.4 ± 2.0 Hz and 0.25 ± 0.15 s, respectively, Fig 1B) and extended over the reward delivery period (liquid rewards were delivered three times at 0.41, 0.82, and 1.23 s after the first lick, thus 0.66, 1.07 and 1.48 s after the cue on average). In contrast, the lick rate in FA trials was also high in the early period (average and standard deviation of lick rate and lick latency, 3.2 ± 2.1 Hz and 0.31 ± 0.23 s, respectively), but it gradually reduced to baseline because no reward was given for No-go cues. We fitted the Q-learning model to the licking behavior of individual mice (n = 17) in 26,517 trials. Fitting performance was good for both Go and No-go trials, which showed an increase in fraction correct and a decrease in fraction incorrect, respectively, at both the population (Fig 1C) and individual (Fig 1D) levels (average and standard deviation of coefficients of determination of 17 mice, 0.87 ± 0.15 for Go cue and 0.61 ± 0.18 for No-go cue, respectively, see Methods for details). The hyper-parameters estimated for individual mice were broadly distributed (average and standard deviation; 0.002 ± 0.002 for $\alpha$, 0.12 ± 0.07 for $q_1$, 0.24 ± 0.17 for $q_2$, 0.14 ± 0.07 for $\tau$, and 0.84 ± 0.24 for $\xi$, Fig 1E), indicating that each animal utilized a distinct strategy for optimally learning to obtain the reward. As discrimination learning progressed, the reward-related variables evolved over time. Initially, Q values for both cues and lick ($s$=Go/ No-go and $a$=Lick) were positive and intermediate. Over time, they increased for HIT trials ($s$=Go and $a$=Lick) and decreased toward negative values for FA trials ($s$=No-go and $a$=Lick, Fig 1F). Note that the negative Q value for FA trials at the later stage of learning accurately reflected the negative R-P value assigned for those trials by the Q-learning model. As

a consequence, $\delta Q$ values converge to zero for both cue-response conditions (Fig 1G). More specifically, the $\delta Q$ values for HIT trials ($s$=Go and $a$=Lick) were positive initially and monotonically decreased during learning. By contrast, the initial $\delta Q$ value for FA trials ($s$=No-go and $a$=Lick) was negative and large, because of a large difference in negative reward R-P and the initial positive value of reward prediction $q_2 > 0$. Throughout the course of learning, this $\delta Q$ value monotonically increased (decreased its magnitude) to zero, indicating a better agreement between negative R-P and Q values. For CR and MISS trials, both Q and $\delta Q$ remained constant at zero (Fig 1F, G).

## Zonal complex spike firings and their correlations with reinforcement-learning and sensorimotor-control explanatory variables

It is important to highlight that the experimental data analyzed in this study is identical to that of the previous study [26]. For clarity, we provide a brief summary of climbing fiber activity here and refer the reader to the previous paper for further details. While mice learned Go/No-go discrimination tasks, we conducted two-photon recordings of climbing fibers (sampling rate, 7.8 Hz) from 6,445 Purkinje cells in eight cerebellar zones (from 7+ to 4b-, see Methods for details). A hyper-resolution algorithm (HA_time [37]) was applied to estimate timing of complex spikes (CSs) at a resolution of 100 Hz. Similar to the previous work, we studied CS responses as population peri-stimulus time histograms (PSTHs) sampled in three learning stages (from top to bottom, 1st, 2nd, and 3rd stages with fraction correct <0.6, 0.6-0.8, >0.8, respectively, Fig 2A) for the four cue-response conditions, or the corresponding four state-action combinations. CS responses in HIT trials (n = 3,788) were initially widespread across the medial zones and prominent in the 5+ and 5a+ zones at the early learning stage. As learning progressed (2nd and 3rd stages), these responses intensified and became more compartmentalized, focusing on positive zones (6+, 5+, and 5a+) in the entire hemisphere. In contrast, CS responses in FA trials (n = 1,757) were distributed across almost the entire hemisphere and gradually decreased and were more confined to lateral zones along with learning. CS responses for CR trials (n = 2,229) were primarily localized within zones 6- and 6+ and showed a slight increase as learning progressed. There was only spontaneous CS activity in MISS trials (n = 201).

In the present study, we systematically studied correlations between the firing activity of neurons in all eight cerebellar zones with rewards, as well as sensorimotor variables. For this purpose, neuronal activity in each trial was defined as the mean firing rate in [-0.5, 2] s after cue onset of neurons in the same aldolase-C zone. For explanatory variables, we included R, Q, and $\delta Q$ as reward variables, lick-latency fluctuation and lick count in the three response windows ([0, 0.5] s for early lick – ELick, [0.5, 2] s for reward lick – RLick and [2,4] s for late lick – LLick) for Go and No-go cues, separately, as sensorimotor variables. Here, we prepared 6 sensorimotor variables related to lick count ($2 \times 3 = 6$; Go vs No-go multiplied by three response windows of licks) (Fig 2B, inset). Note that these three response windows correspond well with licking behavior of mice, as well as reward delivery period (Fig 1B). We further note that the physical reward R (R = 1 for HIT trials and R = 0 otherwise), which is used as one of the explanatory variables for the regression analysis later, is different from the reward-punishment R-P used in the Q-learning model (see Methods for details).

The correlation analysis was performed using partial least squares (PLS) regression to identify meaningful relationships even in the presence of multicollinearity among the explanatory variables. The correlations between these variables ranged from 0.12 ± 0.23, with the highest correlations observed between R and Q (0.82) and R and $\delta Q$ (0.78). We calculated the variable importance in projection scores (VIP scores) to quantitatively estimate the importance of a

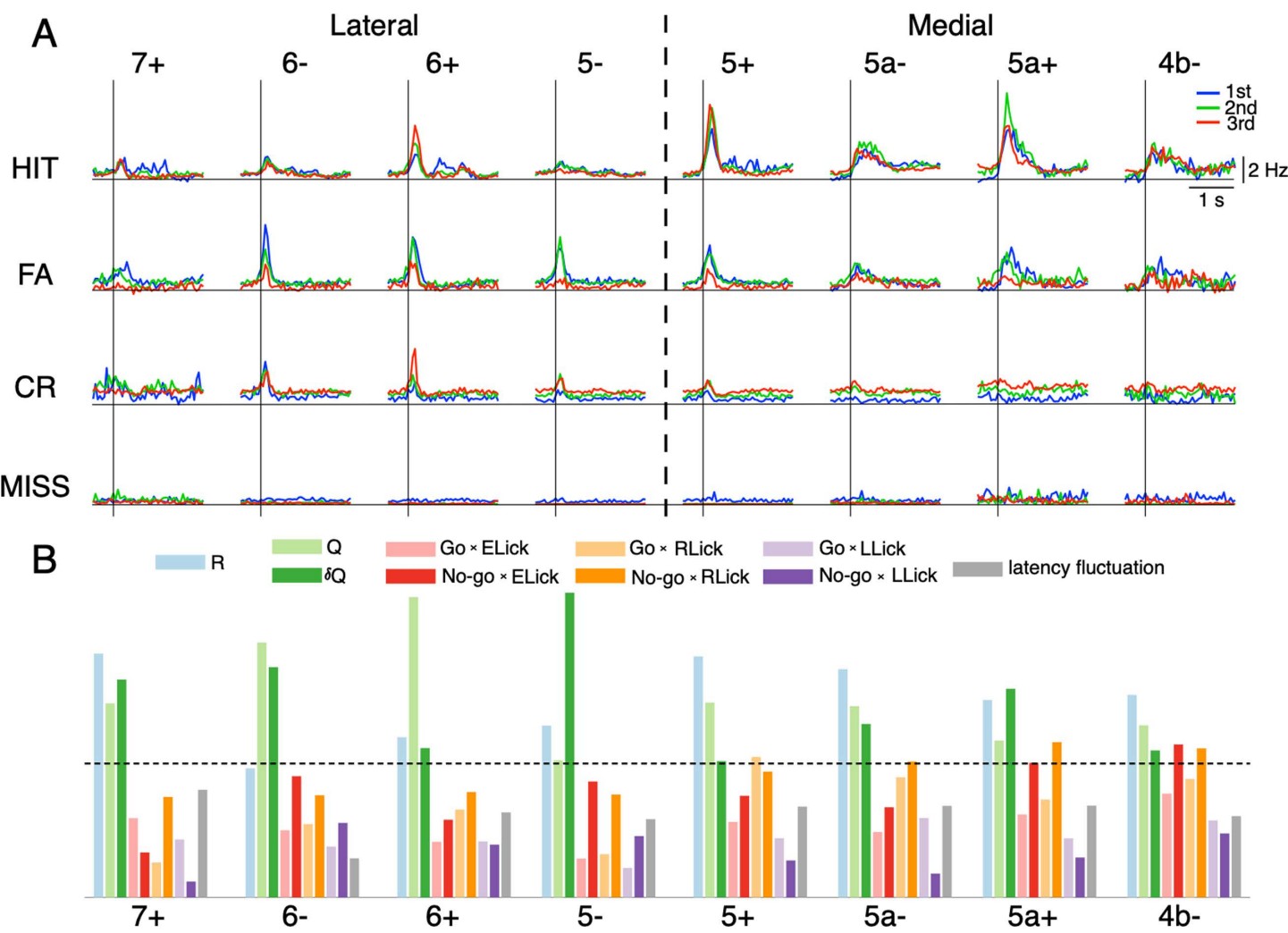

**Fig 2. CS activities to cues and their correlations with reward and sensorimotor variables.** A: Panels showed PSTHs of CSs in 8 aldolase-C zones (7+ to 4b-, columns) in the four cue-response conditions (rows). Blue, green and red traces are for 1st, 2nd and 3rd learning stages, respectively. The horizontal lines indicate cue onset. Dashed horizontal lines represent the boundary between lateral vs. medial parts of the left Crus II. B: Bars showed the variable-importance-in-prediction (VIP) scores of 10 reinforcement-learning and sensorimotor-control variables (from left to right, R, Q, δQ, Go × ELick, No-go × ELick, Go × RLick, No-go × RLick, Go × LLick, No-go × LLick and latency fluctuation) for spiking activity of neurons in 8 aldolase-C zones. Dashed lines indicated VIP score = 1, which is considered a threshold of importance. See the inset for color codes of the 10 explanatory variables.

given explanatory variable to the PLS regression model. The VIP score does not provide the sign (positive or negative) of correlations, but explanatory variables, whose VIP scores are larger than 1, are generally considered important in PLS regression (see Methods for details). We found that neurons in the lateral zones were strongly associated with Q and δQ, while only those in 7+ were linked to the reward R (VIP score = 1.8, Fig 2B). More specifically, 6- and 6+ were strongly associated with Q (VIP score, 1.9 and 2.2 for 6- and 6+, respectively), while 6- and 5- were strongly associated with δQ (VIP score = 1.7 and 2.3 for 6- and 5-, respectively). In contrast, neurons in the medial zones (5+ to 4b-) were strongly associated with the reward R (VIP score > 1.4), and the lick number in the reward period (Go × RLick and No-go × RLick, VIP score > 1). Only 5a+ and 4b- were associated with early licks following No-go cues (No-go × ELick, VIP score >1). No zones exhibited correlations with late lick count and latency fluctuation (VIP score < 0.8).

## The generative model of spiking activity at a trial basis by tensor component analysis

The PLS regression shown in Fig 2 suggested a functional organization of CSs, moderately constrained by the zonal structure, with respect to sensorimotor and reward processing. In our previous study, we conducted tensor component analysis (TCA) for CS responses of >6,000 PCs and found four well-separated components that explained more than 50% of variance in PSTHs (see [26] for details). Remarkably, each zone and even each neuron contained multiple functional components, as supported by previous studies demonstrating multiplexed representations [38,39]. However, the TCA in the previous study was based on averaged CS responses across trials, leaving the activity of functional components in individual trials unknown. To overcome this limitation and disentangle the multiplexed functions of each component, we extended the TCA from the previous averaging study to incorporate trial-based analyses.

In this study, we first applied the same TCA approach as in the previous study to identify four tensor components (TC1-4, Fig 3A). We then used TCs as a generative model of spiking activity as a trial basis to reveal associations of functionally organized CSs and

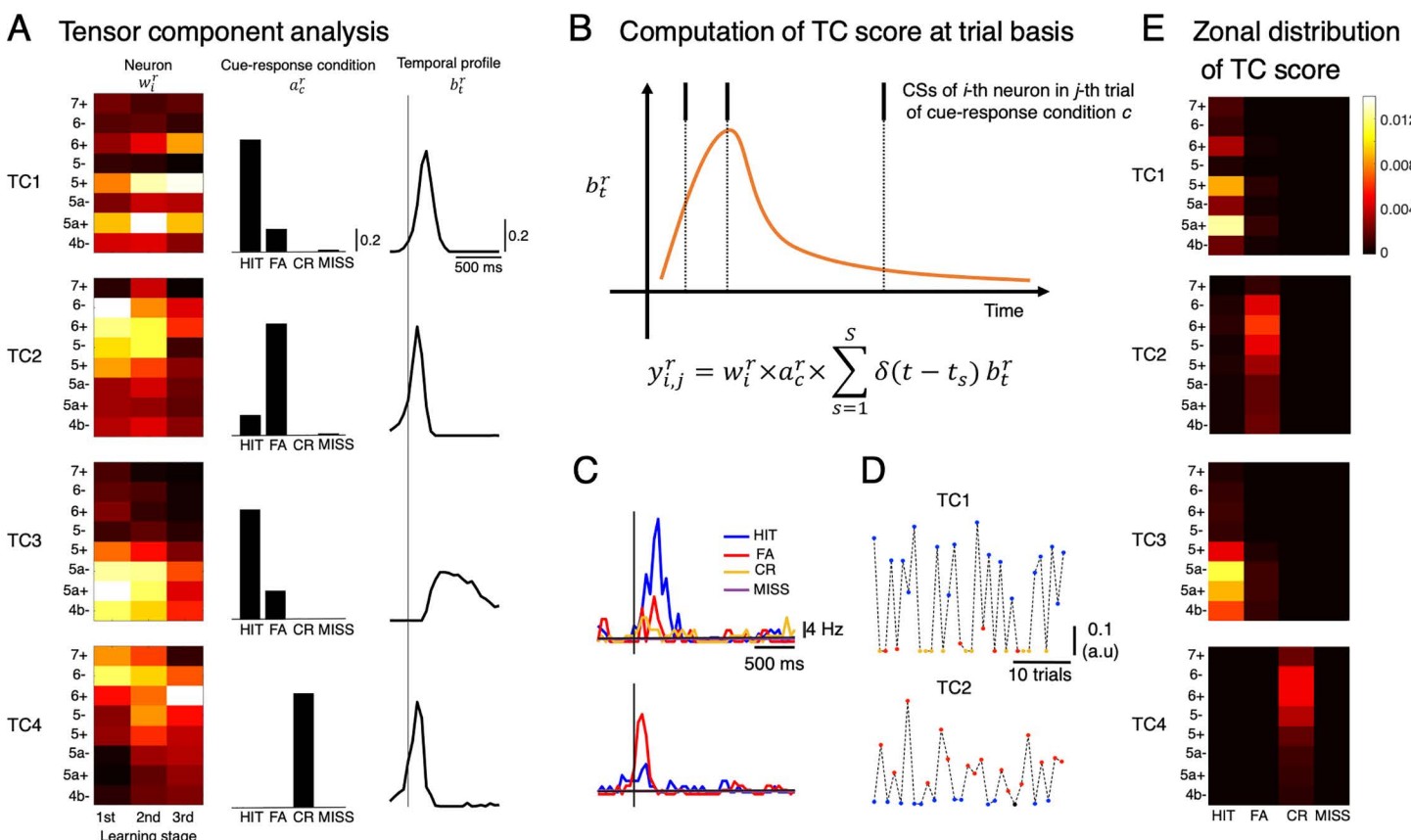

**Fig 3. Tensor-component analysis (TCA) and computation of tensor score at a trial-basis.** A: TCA was conducted for PSTHs in 4 cue-response conditions of n=6,445 neurons and the resulting four tensor components (TC1-4) explained more than 50% of variance. B: for the i-th single neuron, its activity for the r-th TC ($y^r$) in the particular j-th trial was computed by filtering spike timings with temporal profile of the r-th TC $b_t^r$, multiplying corresponding coefficients $w_i^r$ of the i-th neuron and $a_c^r$ of the cue-response condition c. C-D: PSTHs (C) of two representative neurons, which have the highest coefficients of TC1 and TC2, respectively, and their TC1 and TC2 scores, respectively, computed for all trials in their corresponding sessions (D). E: Heatmaps showed TC1-4 scores averaged for all neurons in each of the eight zones distinctively for the four cue-response conditions.

reinforcement-learning and sensorimotor-control variables. Briefly, the TC score of a neuron in a particular trial was estimated by filtering the spike train of that neuron by the temporal profile of the corresponding TC, weighted by neuronal and cue-response condition coefficients (Fig 3B, see Methods for details). This computation incorporated trial-to-trial variability in spiking activity while maintaining fundamental properties of TCs. For example, neurons, whose TC1 and TC2 coefficients were highest among all neurons recorded, had high TC1 and TC2 activities in HIT and FA trials, respectively (Fig 3C-D). Note that TC1 and TC2 were selectively activated in HIT and FA trials, respectively (Fig 3A). Following this computation, the TC score, averaged TC activity of all neurons in the same recording session, shared a similar structure of zonal distribution and cue-response condition with those of the TCs (compare Fig 3A and 3E, but note that the abscissae are learning stages and cue-response conditions, respectively). Specifically, TC1 scores were high in HIT trials and for positive zones. In contrast, TC2 scores were high in FA trials and distributed in the lateral zones. TC3 scores were high in HIT trials and distributed in the medial zones. TC4 scores had similar zonal distribution with TC2 scores, except that they were non-zero only for CR trials.

## Sparse canonical correlation analysis

To find the variables that contribute the most to each TC score, we conducted sparse canonical correlation analysis (sCCA) between the TC scores and the same 10 reinforcement-learning and sensorimotor-control variables used in the PLS regression (see Methods for details). As a result, TC1 and TC3 were associated with reward variables only while TC2 and TC4 were associated with both reward and sensorimotor variables in No-go trials (Fig 4A). More specifically, TC1 and TC3 were positively correlated with reward R, reward prediction Q, and reward prediction error $\delta$Q, with high coefficients of R and Q for TC1 (0.68 and 0.62) and R for TC3 (0.81). We can safely state that TC1 is mainly related to reward and its prediction, and that TC3 is mainly related to reward. Remarkably, TC2 was negatively correlated with $\delta$Q (coefficient, -0.85) but it was positively correlated with the early lick count in No-go trials (0.41). Similarly, TC4 was negatively correlated with both R (coefficient, -0.67) and early lick count in No-go trials (-0.59). In agreement with the previous study [26], TC1 showed a negative correlation with latency fluctuation, albeit with a relatively small coefficient (-0.06). We observed that these associations can also be identified in individual animals (S1A Fig) and in neurons predominantly representing the four TCs (S1B Fig). We further confirmed the significant correlation for each TC component by linear regression of TC1 vs. Q (slope = 0.46, p < 0.0001, Fig 4B), TC1 vs. $\delta$Q (slope = 0.35, p < 0.0001, Fig 4C), TC2 vs. $\delta$Q (slope = -0.42, p < 0.0001, Fig 4D), TC3 vs. R (slope = 0.30, p < 0.0001, Fig 4E) and TC4 vs. No-go $\times$ ELick (slope = -0.24, p < 0.0001, Fig 4F). Notably, these correlations were also significant (p < 0.0001) with comparable slopes, even when using only trials of cue-response condition with which each TC is primarily associated (slope = 0.32 for TC1-HIT, slope = -1.24 for TC2-FA, and slope = -0.36 for TC4-CR trials, Figs 4B, D, F). The most interesting exception was obtained for TC1 vs. $\delta$Q for HIT trials. The correlation was strongly significant (slope = -0.65, p < 0.0001, Fig 4C), but the correlation was negative for only HIT data (slope = -0.65), while the overall correlation was positive as shown above (slope = 0.35).

## Spiking neural network model of modular reinforcement-learning in Go/No-go tasks

Based on observed significant differences in CF responses between HIT and FA trials (Figs 2A and S2), and the cue-specific nature of TCs (Fig 3A and E), we propose that Crus II operates within a modular framework: TC1 and TC3 for Go cues and TC2 and TC4 for No-go cues.

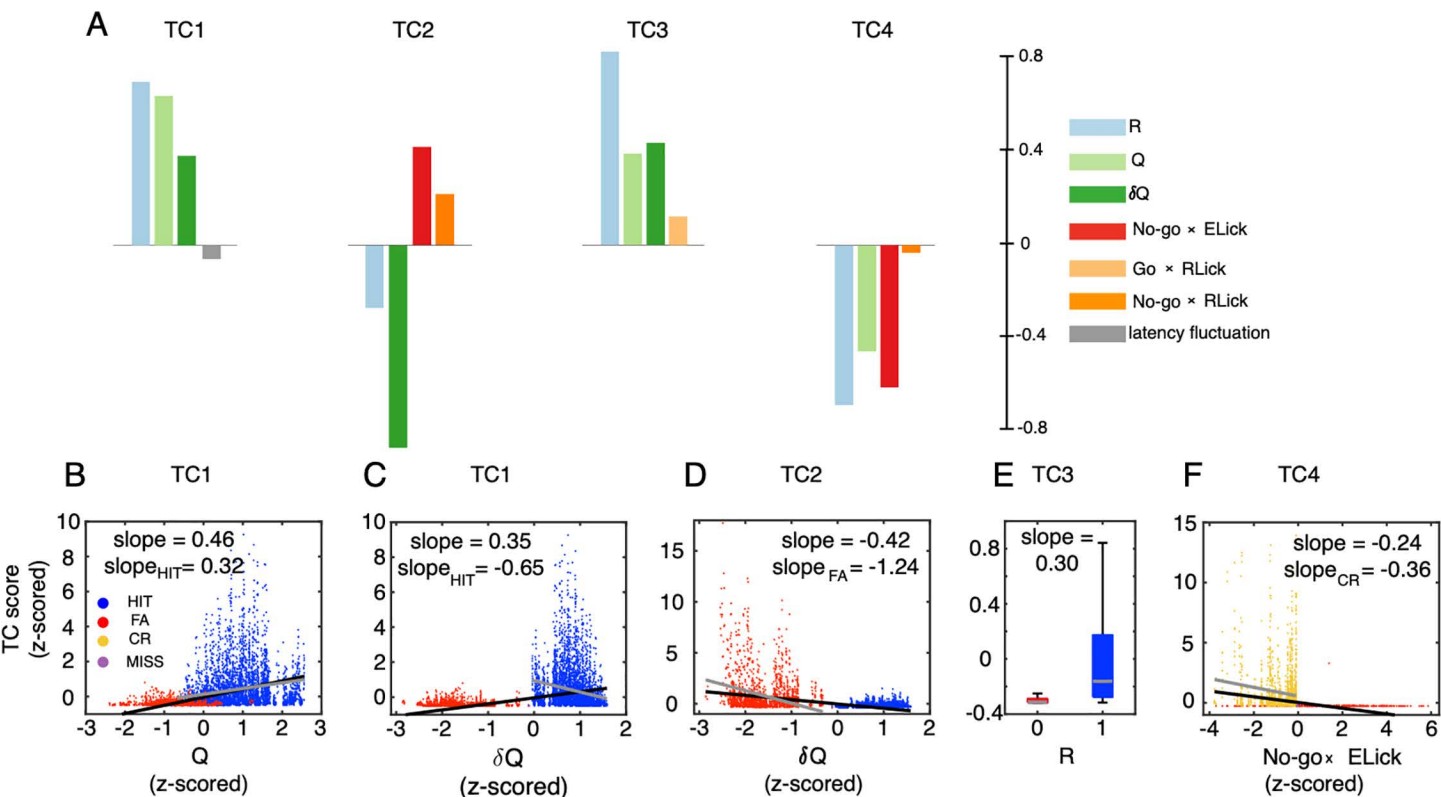

**Fig 4. Sparse canonical-correlation analysis (sCCA) of TC scores with reinforcement-learning and sensorimotor-control variables.** A: Bars show coefficients of reinforcement-learning and sensorimotor-control variables corresponding to TC1-4 scores. B-F: the scatter plots of trials showing correlations of TC1 with Q (B), TC1 with δQ (C), TC2 with δQ (D), TC3 with R (E) and TC4 with No-go × ELick (F). Black and gray lines indicate regression between variables when using all trials and trials of the cue-response condition with which each TC is primarily associated, i.e., TC1-HIT, TC2-FA and TC4-CR, respectively. Panel E shows the boxplot with gray lines indicating the median and the bottom and top edges of the box the 25th and 75th percentiles, respectively. All correlations in B-F are significant (p < 0.0001). Color convention of trials is the same as Fig 1. The inset of A shows color codes of the selected 7 reward and sensorimotor variables among 10 according to sCCA.

Furthermore, CF inputs in TC1 and TC2 exhibit negative correlations with reward prediction errors for Go and No-go cues, respectively (Fig 4C and D), corroborated by statistical analysis (S3 Fig). Together, these findings suggest that Purkinje cells function as context-specific actors, generating essential motor commands based on reward-prediction errors relayed by climbing fiber inputs. This framework integrates cerebellar supervised learning with modular reinforcement learning as follows.

According to our hypothesis, two types of neuronal populations develop necessary motor commands through reward prediction errors transmitted by their climbing fibers for Go and No-go cues, separately (Fig 5A). During initial learning stages, substantial negative reward prediction errors strongly activate CSs of the No-go cue population containing TC2 neurons. If simple spikes (SSs) of these neurons also fire within 0-500 ms at a baseline rate of 50-100 Hz, this leads to numerous inappropriate licks. The concurrent activation of parallel fiber and climbing fiber inputs at this stage likely induces long-term depression (LTD) at parallel-fiber-to-Purkinje-cell (PF-PC) synapses, reducing SS modulation and mirroring CS modulation. This is consistent with simulations by [40] regarding adaptive control of ocular following responses. As learning progresses, negative reward prediction errors diminish, and CSs activate less, with SSs of these neurons facilitating suppression of licks following No-go cues. In this scenario, TC2 neurons turn into TC4 neurons within the population for No-go

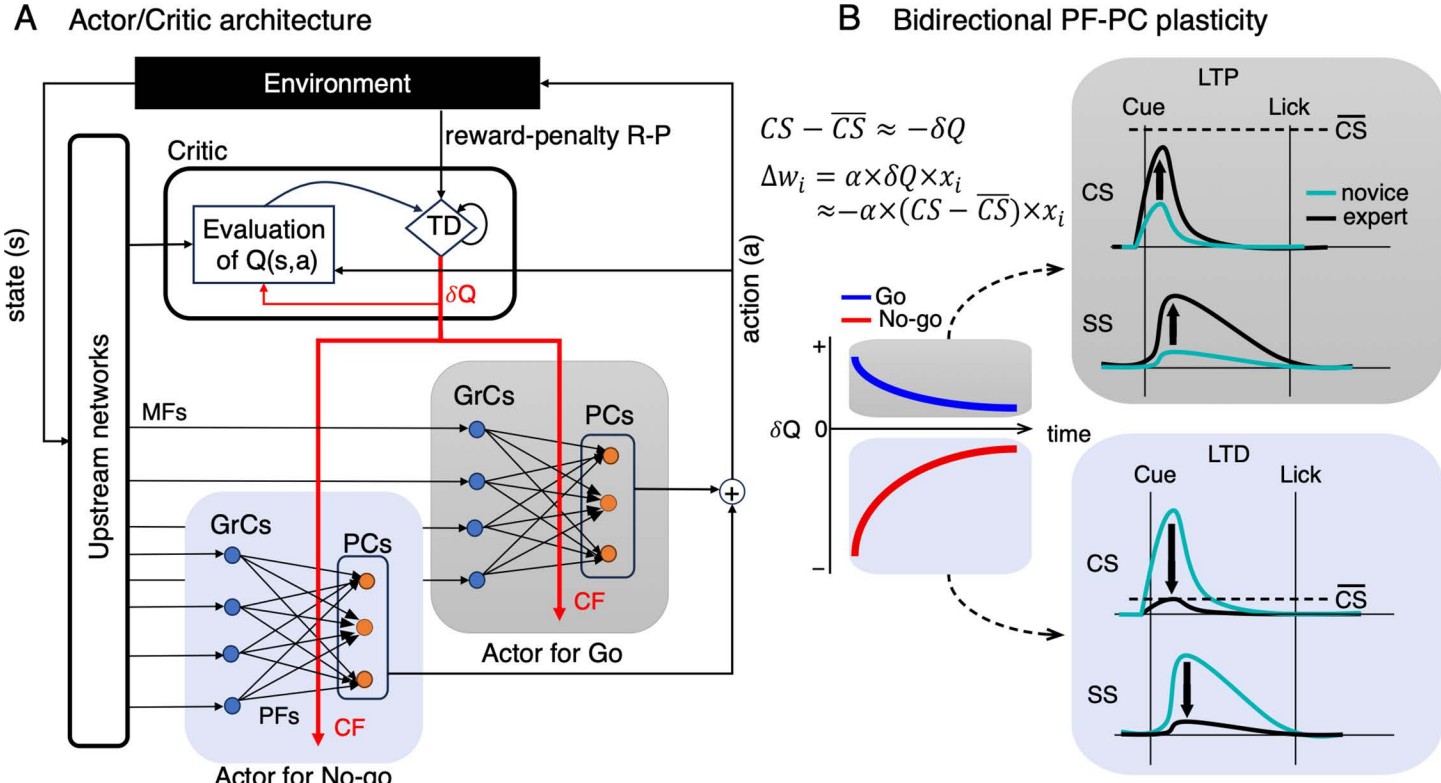

**Fig 5. The conceptual model for modular reinforcement learning of the Go/No-go task and bidirectional plasticity at the PF-PC synapses.** A: The state (*s*) and action (*a*) are conveyed to the critic by upstream networks and efference copies from the actors, respectively. The critic then computes a temporal-difference (TD) prediction error δQ as comparing the observed reward-penalty value R-P with the predicted Q value. The prediction error δQ is used to update the state-action dependent reward prediction in the critic as well as policy of the actors (red arrows). In the Go/No-go task, subsets of Purkinje cells act as context-dependent actors for Go (gray shade) and No-go (blue shade) cues separately. Here, we postulated that two neuronal populations acquire necessary motor commands by utilizing negative reward-prediction error δQ, relayed by their CF inputs, in a supervised learning framework. B: bidirectional PF-PC plasticity may occur depending on the magnitude of CSs. Consequently, modulation of SSs was in the same direction with change of CS activities during learning (black arrows). Note that CSs of TC1 and TC2 neurons were negatively correlated with reward-prediction errors in Go (blue line) and No-go (red line) trials, respectively. Horizontal dashed lines indicate the threshold $\overline{CS}$, which determines LTD or LTP at the PF-PC synapses. MFs – mossy fibers, PFs – parallel fibers, CF – climbing fiber, GrCs – Granule cells, PCs – Purkinje cells, LTP – long-term potentiation, LTD – long-term depression, CS – complex spike, SS – simple spike.

cue as suggested in [26]. Therefore, TC4 and TC2 may share the same underlying neural network and mechanisms of an actor for No-go cue. In contrast, TC1 neurons within the population for Go-cue show lower CS activity early on (S2 Fig), indicating a tendency for long-term potentiation (LTP) at their PF-PC synapses. Thus, their SSs likely increase over time, enabling more precise licking responses to Go cues, acting as an actor for Go cue. Likewise, the neural network underlying TC3 may resemble that of TC1, with the key distinction being that TC3 was specifically activated during the reward delivery period (S2 Fig). This suggests that TC3's climbing fibers transmit distinct or multiplexed information to modulate licking behaviors during this phase. It's worth noting that synaptic plasticity could be also zonal-dependent, with TC1 neurons, primarily distributed in aldolase-C positive zones (Fig 3A), are more likely to undergo LTP [41–44]. Intriguingly, the bidirectional plasticity at PF-PC synapses, potentially driven by climbing fibers, seems to follow a unified learning rule (Fig 5B):

$$\Delta w_i = \alpha \times \delta Q \times x_i \approx -\alpha \times \left(CS - \overline{CS}\right) \times x_i$$

Here, $\alpha$, $x_i$, and $w_i$ represent the constant learning rate, PF input, and PF-PC synaptic weight of the $i$-th granule cell, respectively. The parameter $\overline{CS}$ denotes the threshold amplitude determining plasticity, with LTP requiring a larger value than LTD [45]. This learning rule suggests that LTP (positive $\Delta w_i$, synaptic weight increases) occurs when $CS - \overline{CS} < 0$, whereas LTD (negative $\Delta w_i$, synaptic weight decreases) happens when $CS - \overline{CS} > 0$ (Fig 5B). As a proof-of-concept, we next developed a cerebellar spiking neural-network model featuring a modular framework for reinforcement learning (Fig 6). Based on reasonable assumptions including the bidirectional PF-PC plasticity, this model will successfully replicate both the licking behaviors and the firing rates of climbing fibers observed in Go/No-go tasks (Fig 7).

To objectively examine the conceptual model proposed in Fig 5 by simulation, we developed a neural network model of TC1-4 neurons with 5,000 spiking neurons within a modular reinforcement learning framework engaged in Go/No-go tasks. In this model, we made two fundamental assumptions. First, we assumed the presence of two groups of neurons, corresponding to TC1 & TC3 (termed "$TC_{Go}$") and TC2 & TC4 ("$TC_{Nogo}$"), operating

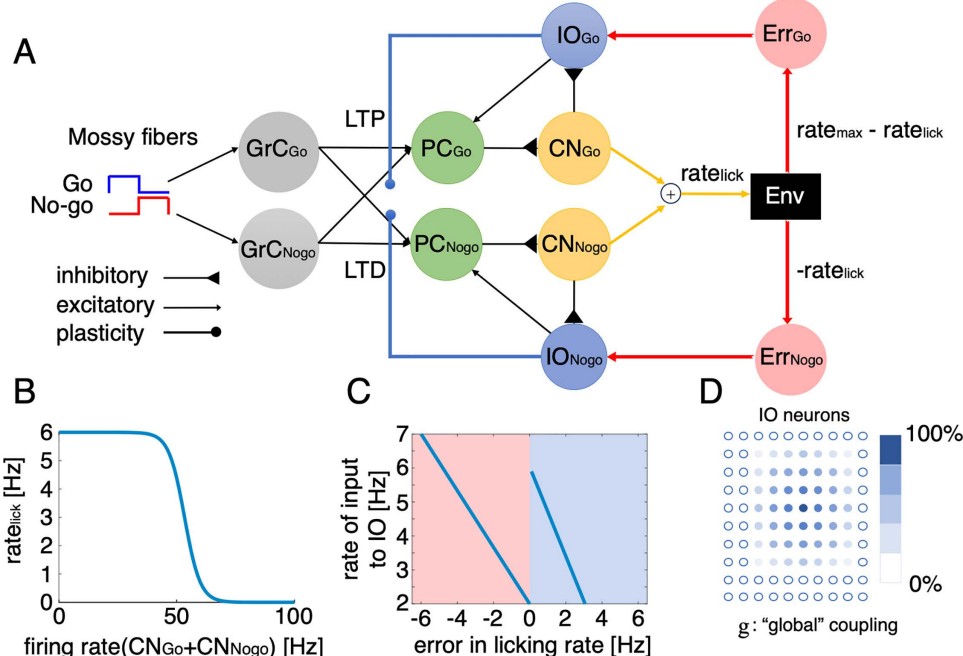

**Fig 6. Spiking neural network model of the cerebellum with 5,000 neurons in Go/No-go tasks.** A: The model consists of two groups of neurons in the PC–CN–IO circuitry, each corresponding to TC1 & TC3 ($TC_{Go}$: $PC_{Go}$–$CN_{Go}$–$IO_{Go}$) and TC2 & TC4 ($TC_{Nogo}$: $PC_{Nogo}$–$CN_{Nogo}$–$IO_{Nogo}$). Sensory input to the $PC_{Go}$ and $PC_{Nogo}$ were transmitted via mossy fibers (MFs) to granule cells for Go ($GrC_{Go}$) and No-go ($GrC_{Nogo}$), respectively. Note that the two neuronal groups received shared mossy fiber input, which is represented by equal connection of $GrC_{Go}$ and $GrC_{Nogo}$ to both $PC_{Go}$ and $PC_{Nogo}$. In this model, LTP and LTD are assumed to occur at PF-PC synapses of $TC_{Go}$ and $TC_{Nogo}$, when IO firing is lower and higher than the threshold, respectively. For each group, PCs, CN, and IO designated by green, yellow and blue discs contained 100 simulated neurons each, and we prepared 2000 GrCs for both Go ($GrC_{Go}$) and No-go ($GrC_{Nogo}$) cues. B: The lick rate is modeled as a sigmoid function of the combined firing rates of $CN_{Go}$ and $CN_{Nogo}$ neurons, with the maximum lick rate ($rate_{max}$) set at 6 Hz. C: The error rates of Go and No-go trials, defined by the difference between the target lick rate ($rate_{max}$ for Go and 0 for No-Go trials) and the actual lick rate, are transformed into the rate of Poisson spike generator inputs $Err_{Go}$ and $Err_{Nogo}$ to $IO_{Go}$ and $IO_{Nogo}$ neurons, respectively. This reproduces the established negative correlations between $\delta Q$ and CSs in Go trials for $TC_{Go}$ (blue region) and No-go trials for $TC_{Nogo}$ (red region). D: A lattice structure with 10x10 IO neurons for each of $TC_{Go}$ and $TC_{Nogo}$ is modeled, where the effective coupling strength between neurons is proportional to their relative distance. In each trial, the effective coupling strength was determined by the firing rate of CN neurons (see Methods for details).

within a modular architecture where each group independently processes Go and No-go cues while sharing the common mossy fiber inputs (Fig 6A). Each group comprises Purkinje cells (PCs), neurons in the cerebellar nuclei (CN), and inferior olive (IO) neurons, with biologically plausible connections forming a closed loop. Briefly, PCs send inhibitory signals to the CN, which, in turn, send inhibitory signals to the IO neurons and effectively regulate their electrical couplings. IO neurons convey reward-prediction error signals to PCs via climbing fibers, completing the loop. In this model, it is assumed that sensory information

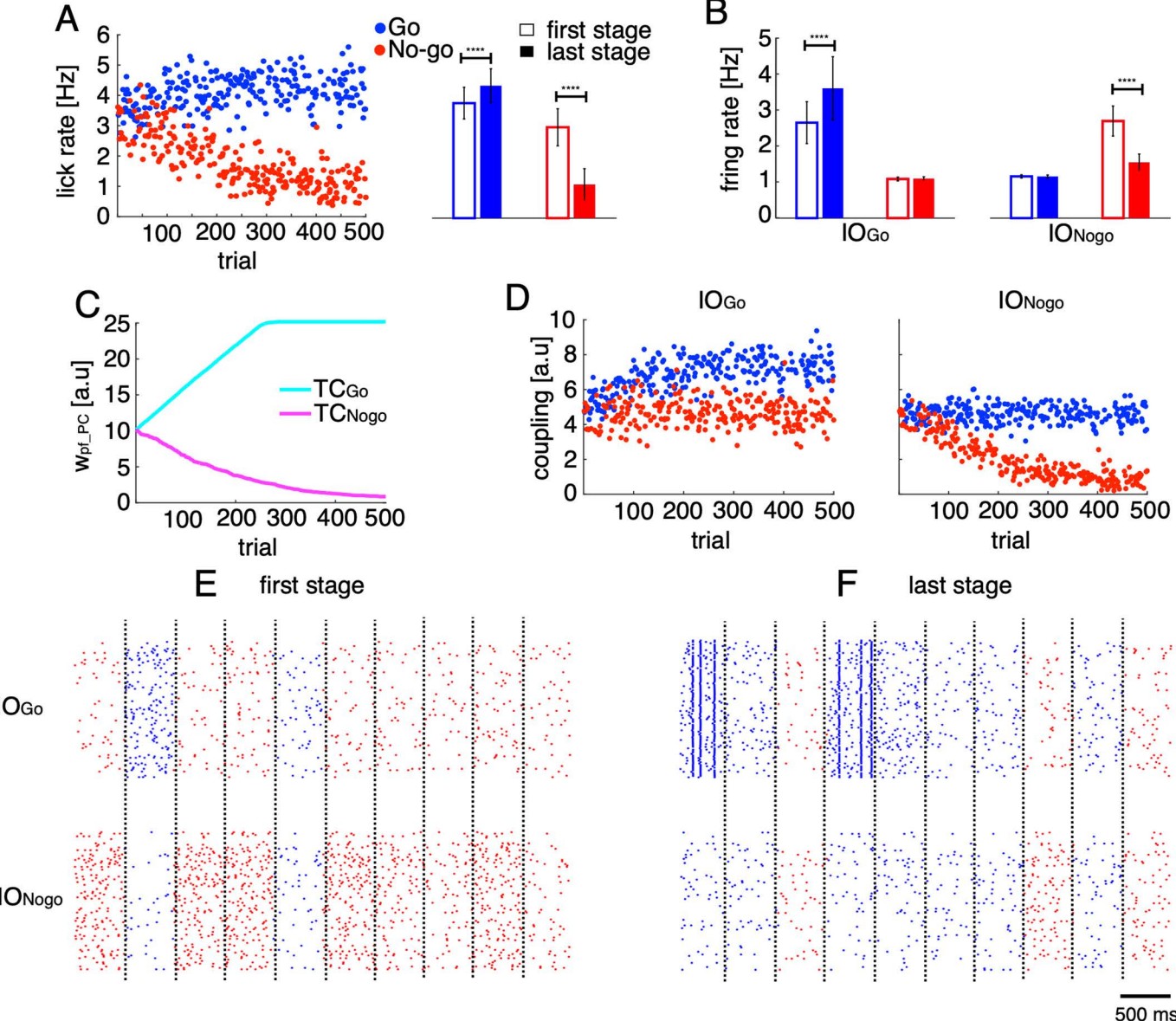

**Fig 7. Licking behaviors and neural firings of the model.** A: lick rate in 500 randomly generated trials, distinguished by Go (blue dots) and No-go (red dots) cues. Each dot represents a single trial. The right panel presents mean ± std of lick rates in the first 100 trials (open bars, first stage) and the last 100 trials (filled bars, last stage). B: firing rates of $IO_{Go}$ and $IO_{Nogo}$ neurons in the first and last stages of the trials. C: bidirectional changes in the weight of PF-PC synapses for $TC_{Go}$ (cyan trace) and $TC_{Nogo}$ (magenta trace) throughout the learning process. D: effective coupling between $IO_{Go}$ (left) and $IO_{Nogo}$ (right) neurons in individual trials. E-F: raster plots of $IO_{Go}$ (upper panel) and $IO_{Nogo}$ (lower panel) neurons in the first (E) and last (F) stages. Vertical dashed lines indicate trial (Go vs No-go) boundaries. Asterisks in A-B indicate significance level of the t-tests between the first and last stages: n.s, p < 0.05; ****, p<0.0001.

of auditory cues is transmitted through mossy fibers to two distinct sets of granule cells, each corresponding to Go and No-go cues. All granule cells provide excitatory inputs to all PCs (see Methods for details). Second, we assumed that PF-PC synapses of $TC_{Go}$ and $TC_{Nogo}$ undergo LTP and LTD, respectively. This bidirectional plasticity was achieved by a single bidirectional synaptic-plasticity formula equally applied to $TC_{Go}$ and $TC_{Nogo}$, where the CS response is compared to a threshold set higher for $TC_{Go}$ than for $TC_{Nogo}$ (refer to Fig 5B for underlying details of this assumption). In addition to these two assumptions, to simulate reward processing in the Go/No-go task, we also assumed that the reward size is piecewise linearly proportional to the lick rate, which is the output of the model. Consequently, the model learns to maximize rewards by increasing the lick rate to a theoretical maximum (set as 6 Hz in the simulation) for Go cues and to minimize penalties (negative rewards) by decreasing the lick rate toward 0 Hz for No-go cues. We modeled the lick rate as a function of the combined firing rates of CN neurons (Fig 6B). The error in licking rate, which is piecewise linearly proportional to reward-prediction error, was computed as the difference between a target lick rate and the actual lick rate output from the model. Finally, this error in lick rate was linearly translated into the input firing rates to IO neurons (Fig 6C), consistent with correlations identified by sCCA (Fig 4C&D). Since neurons in the CN send inhibitory signals to regulate the effective coupling between IO neurons via gap junctions [46–49], and building upon our previous research [26], which reported dynamic changes in climbing fiber synchrony, we also examined how inhibition from the CN modulates the effective coupling between IO neurons (Fig 6D).

It's important to emphasize that due to a large gap between the numerous model parameters and the available real data, our aim was not to fit the model to the data precisely. Instead, the model is designed to qualitatively replicate licking behaviors and CS firings observed in Go/No-go tasks (see Methods for details) to conceptually support our hypothesis on modular reinforcement learning system implemented by supervised learning. In this simulation, behaviorally, the lick rate following Go cues increased from 3.7 ± 0.5 Hz in the first 100 trials (first stage, Fig 7A) to 4.2 ± 0.5 Hz in the last 100 trials (last stage, t-test between two stages, $p < 0.00001$). In contrast, during the same learning period, the lick rate following No-go cues reduced by almost threefold (2.9 ± 0.6 and 1.1 ± 0.5 Hz for the first and last stages, respectively, $p < 0.00001$). Note that with these changes in licking behaviors, the error rate for both cues gradually approached zero during the learning process. Regarding the neural activity in simulation, the IO firing rates increased for $TC_{Go}$ (2.7 ± 0.6 and 3.7 ± 0.9 Hz for the first and last stages, respectively, $p < 0.00001$, Fig 7B) and decreased for $TC_{Nogo}$ (2.5 ± 0.4 and 1.4 ± 0.2 Hz for the first and last stages, respectively, $p < 0.00001$). The IO firing rates of $TC_{Go}$ and $TC_{Nogo}$ in No-go and Go trials, respectively, remained unchanged at the baseline level around ~1 Hz (see S4 Fig for firing activity of the neurons in the PC-CN-IO circuitry). It is worth noting that the lick rate and firing rate of IO neurons generated by the model closely matched with those observed in the real data (see Figs 7A-B and S5A-B for comparison). Due to the changes in IO firing, the weights at PF-PC synapses were bidirectionally altered from an initial value of 10 to 25 for $TC_{Go}$ (from $GrC_{Go}$ to $PC_{Go}$) and to nearly zero for $TC_{Nogo}$ (from $GrC_{Nogo}$ to $PC_{Nogo}$, Fig 7C). Similarly, the effective coupling strength between IO neurons also bidirectionally changed, due to bidirectional changes in CN inhibition, from an initial value of 5 to 8 for $TC_{Go}$ in Go trials and to nearly zero for $TC_{Nogo}$ in No-go trials (Fig 7D). As a result, the synchronization of IO firing increased for $TC_{Go}$ in Go trials and decreased for $TC_{Nogo}$ in No-go trials (Fig 7E-F), consistent with findings in our previous study [26]. Notably, strong synchronized IO firing of $TC_{Go}$ were observed in Go trials at the last stage of learning, which might be useful for precise timing control, a phenomenon illustrated by the negative correlation between TC1 and lick-latency fluctuation (Fig 4A).

## Discussion

In this study, we conducted two correlation analyses to elucidate contributions of CSs to licking behavior during learning of a Go/No-go auditory-discrimination task. In the first analysis, building upon distinct contributions of zonal-organized CF inputs to behavioral variables [25], we examined the firing rate of neurons across all eight cerebellar zones against 10 reinforcement-learning and sensorimotor-control variables, using partial least-squares regression to handle multicollinearity. This analysis indicated a moderate distribution of variables across zones, reflecting the multifunctional nature of each zone and neuron [26,38,39]. The second analysis used tensor component analysis to decompose spiking activity into four tensor-components, examining their trial-based functional representations. We employed different complex spike measures across the analyses, ranging from mean firing rates to precise spike timings. Our methods evolved from focusing on eight zones to four tensor components, and from basic linear regression to advanced techniques like sparse canonical correlation analysis. Despite these methodological differences, our findings were consistent. We found that TC1 neurons, distributed most densely in 6+, were positively correlated with reward R, reward prediction Q, and reward prediction error δQ (Fig 2B (6+), Fig 4A&B). However, we note that TC1 is negatively correlated with δQ when constrained to HIT trials (Fig 4C). In contrast, TC2 neurons in zones 6- and 5- were negatively correlated with reward prediction error δQ (Fig 2B(6- and 5-) and Fig 4D). TC3 neurons in medial zones were positively correlated with reward R and licking in reward delivery period (Fig 2B(5+,5a-,5a+,4b-) and Fig 4E). Finally, TC4 neurons, distributed most densely in 6-, were negatively correlated with both reward R and early licking following No-go cues (Fig 2B (6-) and Fig 4A and F).

Notably, these results were also in good agreement with our previous work. In [26], we found that TC1-4 corresponds to timing control of the first lick, cognitive error signals, reward-related signals and action inhibition, respectively. For TC1, both studies showed a negative correlation with lick-latency fluctuation, though this correlation is weaker in the current study compared to the correlations between TC1 and reward variables. We also show that TC2 was negatively correlated with reward prediction error δQ; thus, cognitive error signals with a positive sign can be computed as sign-reversed reward prediction errors with a negative sign by reinforcement learning algorithms. That is, during learning, cognitive error signals decrease due to an increase (with the same magnitude) of negative reward prediction errors. Both studies corroborate TC3's correlations with reward R and reward licking behavior. Similarly, our previous work indicated TC4 neurons inhibit licks in No-go trials. This is reinforced by sCCA's findings of TC4's negative correlation with No-go ✕ ELick. Furthermore, in line with our previous study that focused on synchronized spikes (i.e., co-activated in time bins of 30 ms), in a supplementary analysis, we used synchronized spikes instead of all spikes to compute the TC scores. The sCCA analysis revealed that while the correlations of TC1-4 with other variables remained unchanged, significant negative correlations of TC1-3 and latency fluctuation emerged (S6 Fig). This result strongly aligns with the findings reported in the previous study (see Figs 5 - S1 in [26]). Both studies, therefore, suggest that synchronization of CSs is crucial for timing control [50–52]. We note that in the previous study [26], we identified correlations between TC2 and the "error" response–defined as licks in the 0–0.5 s window after No-go cues– and between TC3 and the "reward" response–defined as licks in the 0.5–2 s window after Go cues. In this study, these responses were redefined as two s ensorimotor-control variables No-go ✕ ELick and Go ✕ RLick, respectively. This new nomenclature more effectively distinguishes reward-related variables from sensorimotor-control ones, and also facilitates the systematic regression analyses performed here.

## Modular reinforcement learning characteristics

In an additional investigation, our objective was to distinguish the roles of complex spikes (CSs) in cue-cognition from those in motor functions. We introduced an auditory cue variable (Cue = 1 for Go, Cue = 0 for No-go) distinct from motor-related variables (i.e., lick counts). This revealed that the correlations of TC1-4 with motor-related variables were dramatically suppressed, while strong correlations with auditory cues emerged (S7 Fig). These findings, supported by previous research [21,22], indicate that no single TC exclusively encodes basic licking motor commands without considering auditory cues. Thus, the cerebellar reinforcement learning algorithm investigated in this study functions as a modular system: TC1 and TC3 for Go cues and TC2 and TC4 for No-go cues, rather than as a simplistic, straightforward algorithm [18,53–57].

We propose that in this task, the acquisition of reinforcement learning Critic, crucial for calculating reward prediction error at cue onset, does not take place in Crus II. Should temporal-difference-error learning be active in Crus II, we would expect to observe complex spike (CS) activities correlating with expected liquid reward delivery [0.5, 2] s, showing contrasting patterns between HIT and FA trials during early learning [36,58–65]. However, among the eight zones and four tensor components (TCs) analyzed, only TC3 neurons displayed firing patterns consistent with this temporal condition, but their activity patterns were not contrasting between HIT and FA for that temporal window, and correlation with δQ was relatively weak (Fig 4A). Therefore, we hypothesized that TC1 & TC3 and TC2 & TC4 neurons function as context-specific Actors within a modular reinforcement learning algorithm for Go and No-go cues, respectively. Intriguingly, their activities begin to rise even before the cue (S2 Fig), suggesting the involvement of internal forward models in preemptively calculating reward prediction errors. These forward models likely involve a network loop that includes the inferior olive nucleus, cerebral cortex, basal ganglia, and cerebellum. They operate anticipatorily and collectively support modular reinforcement learning [66–70]. In this framework, each functional module, represented by Purkinje cells, engages in supervised learning designed for specific actors, as depicted in Fig 5A.

## Technical advances and conclusions

In the present study, we applied the Q-learning algorithm to analyze the licking behavior of mice in Go/No-go tasks and to estimate reward variables. In our computational approach, we differentiate between the state value function $V(s)$, dependent on the state $s$ alone, and the state-action value function $Q(s,a)$, reliant on both the state $s$ and action $a$. Two-photon $Ca^{2+}$ imaging data highlighted the dominance of the state-action value function, with significant differences observed in responses (lick) in HIT vs MISS and FA vs CR trials (Fig 2A). We thus employed a state-action Q-value function for expected reward estimation and used the *softmax* function for action selection based on the learned Q values. Our Q-learning model, with only five hyper-parameters, effectively captured individual mice's learning patterns in licking behavior (Fig 1C and D). Statistical analysis confirmed the importance of all five hyperparameters, as models with fewer parameters showed reduced performance (S8 Fig), emphasizing their relevance in our Go/No-go task. Analysis of the estimated hyper-parameters revealed unique learning strategies among individual mice, with variations in learning rates, penalty values, initial Q values, and individual strategies (temperature) for optimal reward acquisition (Fig 1E).

This study introduces an innovative technical approach by using tensor component analysis (TCA) as a generative model for analyzing spiking activity in single trials, despite the complexity of CFs conveying multiple types of information [26,38,39]. Typically,

unsupervised statistical methods like TCA, while effective in decomposing spiking activity into meaningful components, struggle with noise sensitivity, making them less reliable for single-trial analysis. To overcome this, we applied TCA to peristimulus time histograms (PSTHs) of complex spikes (CSs). This approach reduces trial-to-trial variability, enabling us to identify low-dimensional tensor components (TCs) that represent the underlying dynamics of spiking activity. We then used the temporal profiles of these TCs to refine spike activity analysis in individual trials. As a result, the derived TC scores effectively reflect the spiking dynamics corresponding to the four TCs identified by TCA. This method offers a powerful means to analyze CF activities, providing valuable insights into a range of learning behaviors.

We developed a neural network model of the cerebellum engaged in Go/No-go tasks. The model, although simple, incorporated 5,000 neurons with biologically realistic synaptic connections and featured closed loop connections between Purkinje cells, cerebellar nuclei neurons, and inferior olive neurons. It was embedded within a reward-based learning system, where licking behavior— the model's output —was converted into reward-prediction errors, reflecting our experimental observations. It also simulated the bidirectional plasticity potentially occurring at PF-PC synapses, consistent with experimental findings regarding bidirectional changes in CS activity and the zonal characteristics of the cerebellum. It is important to note a few key points about our model. First, the lick rate is represented as a sigmoid function of the combined spike rate of CN neurons. For simplicity, the model assumes that the lick rate increases as CN spike rates decrease, with the maximum lick rate occurring at low CN spike rates (in our simulations, CN neurons exhibit spontaneous firing rates of approximately 25 Hz (S4 Fig), ensuring that CN activity does not drop to zero). This is supported by the evidence that Purkinje cells influence both excitatory and inhibitory neurons within the CN [71]. If the downstream circuitry of Purkinje cells includes substantial inhibitory connections—such as inhibitory CN neurons and motor neurons—learning a specific motor action may primarily occur through increased simple spike (SS) activity or decreased CN activity. This mechanism has been observed in paradigms like vestibulo-ocular reflex adaptation [72] and reflexive whisker protraction [73], as well as in computational models [74]. Aligned with this mechanism, our model includes inhibitory CN neurons that modulate the electrical coupling between IO neurons. However, we note that experimental studies in eyeblink conditioning [75,76] and other cerebellar models [77] have shown that suppression of SSs, leading to increased CN output, drives movement. In this context, the microzone architecture of the cerebellar cortex may influence activity in downstream circuitry, thereby facilitating specific motor functions [43]. Second, in its current form, the model does not account for the convergent and divergent innervations in the positive feedback loop between PC, CN, and IO, a feature that may be vital for the cerebellum to dynamically organize neuron populations and alter behavior [26]. In particular, the convergence of multiple Purkinje cells onto CN neurons is a critical feature of cerebellar anatomy that likely affects the integration of inhibitory signals in the network. This anatomical feature could potentially lead to an increased level of synchrony between different Purkinje cell inputs, modulating the firing patterns of CN neurons in a way that reflects more complex integration of timing control signals [78–80]. Additionally, the convergence could influence how the cerebellum accounts for noise and variability in the output, since multiple Purkinje cells might help refine the signal sent to CN neurons, smoothing out the response [81]. In our model, we simulated only the lick rates, not the timing of licks, and did not account for noise. Therefore, the convergence of multiple PC inputs onto CN neurons was not considered. Finally, refining the model to closely match experimental data could provide deeper insights into the cerebellum's role in reinforcement learning.

In summary, our study employs a computational approach integrating a Q-learning model, a high-resolution spike detection algorithm, tensor component analysis (TCA), and cerebellar neural network model with 5,000 spiking neurons, to explore reward processing in the cerebellum. The results indicate that inputs from climbing fibers to two distinct Purkinje cell ensembles encode predictive reward-prediction errors. This finding is situated within an integrated framework that combines modular reinforcement learning with supervised learning. Our development of a cerebellar spiking neural network model aligned with this framework successfully replicates both the licking behaviors and the firing rates of climbing fibers observed in Go/No-go tasks. The significance of this framework lies in its potential to enhance the development of new reinforcement learning algorithms, which could efficiently master complex tasks in a limited number of trials [18]. It's important to note some limitations: the complex spikes (CSs) we studied were recorded from Purkinje cells in separate sessions, potentially introducing variability. Also, our research did not track the CSs of individual Purkinje cells throughout the learning process. Consequently, the proposed modular reinforcement learning algorithm in the cerebellum needs further validation through continuous monitoring of individual neurons during learning, coupled with causal analysis of neuronal responses and corresponding behavioral changes.

## Methods

### Ethics statement

All experiments were approved by the Animal Experiment Committees of the University of Tokyo (#P08-015) and the University of Yamanashi (#A27-1), and conducted in compliance with national regulations and institutional guidelines. A total of 17 adult male heterozygous Aldoc-tdTomato mice (n = 12) and 5 adult male wild-type mice (Japan SLC, Inc., n = 5) at postnatal days 40–90 were used. The Aldoc-tdTomato mouse line is available upon request from the corresponding authors and is also accessible at the RIKEN BioResource Center (RBRC10927).

### Q-learning model

We adopted a Q-learning model, a reinforcement learning algorithm, to model licking behavior of mice in a Go/No-go experiment. The algorithm works by learning a state-action value function, commonly referred to as the Q-function. The Q-function is a mapping from a state-action pair to a scalar value that represents the expected reward for taking a particular action in a particular state. The Q-function is updated over time based on the observed rewards and transitions to new states.

In the Go/No-go auditory-discrimination licking task, the Q-function of a state $s$ ($s$ = Go/No-go cue) and an action $a$ ($a$ = Lick/ No-lick) were updated at trial $t$ as follows

$$\delta Q_t = R - P_t - Q_t$$

$$Q_{t+1} = Q_t + \alpha \times \delta Q_t$$

where $\delta Q$ is the reward prediction error, $\alpha$ is the learning rate and R-P is the reward-penalty function.

R-P = 1 for $s$ = Go, $a$ = Lick (HIT)
R-P = - $\xi$ for $s$ = No-go, $a$ = Lick (FA)
R-P = 0, otherwise (CR and MISS)

The probability of selecting a given action $a$ in state $s$ is determined by the *softmax* function comparing Q-function values of an action to all others:

$$Prob(s,a) = \frac{\exp\{Q(s,a)/\tau\}}{\sum_a \exp\{Q(s,a)/\tau\}}$$

where $\tau$ is the temperature parameter, representing a trade-off between exploitation and exploration. Because mice underwent pre-training sessions for 3 days, during which they were rewarded by licking after 1 second of both cues, initial Q values for Go and No-go cues should be positive: $Q_0(s=Go, a=Lick) = q_1$ and $Q_0(s=No\text{-}go, a=Lick) = q_2$. In total, the Q-learning model contains 5 hyper-parameters: learning rate $\alpha$ (range, $0.001 \leq \alpha \leq 0.1$), $\xi$ ($0 \leq \xi \leq 1$), temperature $\tau$ ($0.01 \leq \tau \leq 0.5$), initial Q values for Go and No-go cues, $q_1$ and $q_2$ ($0 \leq q_1 \leq 1$, $0 \leq q_2 \leq 1$). We estimated these parameters for individual animals by maximizing the likelihood defined as the sum of $Prob(s,a)$ for all trials.

It is important to emphasize that we fitted the Q-learning model to behavioral data on a trial basis with trials from different sessions concatenated. To evaluate the fit, we computed from the data the fraction correct for Go cues (number of HIT trials/ total number of Go trials) and the fraction incorrect for No-go cues (number of FA trials/ total number of No-go trials) for each session. These values were then compared to the probability $Prob(s,a)$ of the last HIT and FA trials estimated by the model ([Fig 1C, D]). The coefficient of determination ($R^2$) was calculated to measure goodness-of-fit between the data and model probabilities across sessions of individual animals.

$$R^2 = 1 - \frac{SS_{res}}{SS_{tot}}$$

$$SS_{res} = \sum_i (h_i - f_i)^2$$

$$SS_{tot} = \sum_i \left(h_i - \overline{h}\right)^2$$

where $h_i$ and $f_i$ are the probabilities of the data and the model at the $i$-th session, respectively, and $\overline{h}$ is the mean of the data probability across sessions.

## Estimation of complex spike activity from two-photon recordings

Ca signals in Purkinje cell dendrites were evaluated for regions-of-interest (ROIs) of the two-photon images extracted by Suite2p software [82] and manually selected. Spike trains were reconstructed for 6,445 Purkinje cells sampled in 17 mice, using hyperacuity software (HA_time [37]) that detected complex spike (CS) activities for calcium signals of two-photon imaging with a temporal resolution of 100 Hz (see [26] for details).

## Complex spike to cue stimulus

To evaluate the CS of a single neuron to the cue stimulus, we constructed a peri-stimulus time histogram (PSTH) of CSs in [-0.5, 2] s with a time bin of 50 ms for the four cue-response conditions. They include HIT, FA, CR, or MISS, according to licking behavior within a response period of 1 s to the two cues, i.e., correct lick in response to the go cue, unwarranted lick in response to the No-go cue, correct response rejection to the No-go cue, or response failure to the Go cue, respectively. Each PSTH was subtracted from its baseline activity, defined as the mean value of the firing rate in the range of [-2, -1] s before cue onset.

## Partial least-squares regression analysis

We sought to reveal correlations between zonal activity and behavior variables on a single-trial basis (Fig 3). For each trial, neuronal activity was calculated as the mean firing rate of the neurons in the same aldolase-C zone, in [-0.5, 2] s after cue onset. Behavior variables include physical reward R (R=1 for HIT trials, R=0 otherwise), Q, δQ, number of licks in 0–0.5 s for Go and No-go cues (Go × ELick and No-go × ELick), the number of licks in the reward period (0.5–2 s after cue, Go × RLick and No-go × RLick) and the number of licks in the late period (2–4 s after cue, Go × LLick and No-go × LLick). For a consistent analysis with the previous study [26], we also incorporated lick-latency fluctuation as an explanatory variable. Briefly, lick-latency fluctuation for a single trial was calculated as the absolute difference between the lick-latency and the mean lick-latency across trials for individual mice. This measure was computed separately for HIT and FA trials. For MISS and CR trials, where no licks occurred within the response window, the lick-latency fluctuation was set to the mean values observed in HIT and FA trials, respectively.

We conducted partial least-squares (PLS) regression to resolve multi-collinearity of behavioral variables, e.g., between R, Q and δQ. PLS regression searches for a set of low-dimensional components that performs a simultaneous decomposition of dependent and explanatory variables with the constraint that these components explain as much as possible the covariance between the variables. The VIP (Variable Importance in Projection) score is a measure of the importance of each explanatory variable in a PLS regression model, with higher scores indicating greater importance. The VIP score for the *j*-th variable is given as:

$$VIP_j = \sqrt{\frac{\sum_{f=1}^{F} w_{jf}^2 \times SSY_f \times J}{SSY_{total} \times F}}$$

where $w_{jf}$ is the weight of the *j*-th variable and the *f*-th PLS component, $SSY_f$ is the sum of squares of explained variance for the *f*-th PLS component and *J* is the number of explanatory variables (J=10). $SSY_{total}$ is the total sum of squares explained for the dependent variable, and *F* is the total number of PLS components. Explanatory and dependent variables were standardized to have mean zero and standard deviation 1 before performing the analysis. In our analysis, the number of PLS components was optimized by 10-fold cross-validation. Note that the VIP score does not provide sign information (positive or negative) of correlations between explanatory and dependent variables. Although there was no known threshold for a systematic evaluation, VIP score > 1 was typically used as an indicator for a variable to be considered significant. PLS was conducted using the MATLAB function, *plsregress*.

## Tensor component analysis

Let $x_{ntk}$ denote the PSTH of neuron *n* at time step *t* within cue-response condition *k*. TCA yields the decomposition

$$x_{ntk} \approx \check{x}_{ntk} = \sum_{r=1}^{R} \lambda_r w_n^r b_t^r a_k^r$$

where *R* is the number of tensor components, $w_n^r$, $b_t^r$ and $a_k^r$ are the coefficients of the neuron, temporal, and response condition factors, respectively. Those coefficients were scaled to be unit length with the rescaling value $\lambda_r$ for each component *r*. We introduced a non-negative constraint of those coefficients ( $w_n^r \geq 0$ , $b_t^r \geq 0$ and $a_k^r \geq 0$ for all *r*, *n*, *t* and *k*). In the previous study, we optimized the number of components R = 4 for which solutions were most stable and fitting scores were high [26].

## Computation of tensor component score for a single trial

TCA was efficient for decomposing PSTHs into biologically-meaningful components [26], but for systematic analysis of associations of CSs and variables on a trial basis, it is crucial to estimate activity of such components for individual neurons in a single trial. Note that TCA was carried out for PSTHs computed over multiple trials in a given session, but we needed a firing index for each trial for Q-learning analysis. For that purpose, we employed a novel approach to utilize time series of multiple spikes decomposed by TCA as generative models of CSs.

The tensor-based activity of the $i$-th neuron in the $j$-th trial (corresponding to the cue-response condition $c$) with respect to the component $r$-th ($r = 1,.., 4$) was evaluated as:

$$y_{i,j}^r = w_i^r \times a_c^r \times \sum_{s=1}^{S} \delta(t - t_s) b_t^r$$

where $\sum_{s=1}^{S} \delta(t - t_s)$ represents the timing of CS firings sampled from [-0.5, 2] s after cue onset as the summation of Dirac delta functions $\delta$.

For the $j$-th trial, the TC score of the $r$-th component was calculated as the averaged $y_{i,j}^r$ across all neurons in the same recording session. As a result, each trial has four TC scores corresponding to the four TCs.

$$TCscore_j^r = \frac{1}{N} \sum_{i=1}^{N} y_{i,j}^r$$

## Sparse canonical correlation analysis

Because CSs may multiplex different information, we conducted sparse Canonical Correlation Analysis (sCCA) to analyze the relationship between TC scores and behavior variables. The goal of sCCA is to find a set of linear combinations (known as "canonical variate") of the variables in each set, such that the correlation between the two sets of canonical variates is maximized. The sCCA includes a sparsity constraint, which promotes solutions in which only a small number of variables are selected in calculation of the canonical variates. This can result in more interpretable and biologically relevant solutions, as it reduces the amount of noise in the data.

In our analysis, sCCA was conducted using the *PMA* package of R, with a LASSO penalty applied to enforce sparsity. L1 bounds were set 0.4 and 0.6 for TC scores and behavior variables, respectively (larger L1 bound corresponds to less penalization). Explanatory and dependent variables were standardized to have mean zero and standard deviation 1 before performing the analysis. The number of canonical variables was 4. The resulting coefficient of TC scores was either 1 or -1. For better interpretation, coefficient vectors of behavioral variables (reported in Fig 4A) were re-signed so that coefficients of TC scores were all 1.

## Simulation of the cerebellum in Go/No-go tasks

We developed a spiking neural network model of TC1-4 neurons in a modular reinforcement learning framework for Go/No-go tasks, to illustrate how they generate motor commands to maximize rewards. Our model features two groups of neurons, $TC_{Go}$ (TC1 & TC3) and $TC_{Nogo}$ (TC2 & TC4), operating independently to process Go and No-go cues. We also incorporated bidirectional plasticity in PF-PC synapses for $TC_{Go}$ and $TC_{Nogo}$, reflecting aldolase C+ zones linked with TC1 favoring long-term potentiation (LTP), and aldolase C- zones linked with TC2 being more prone to long-term depression (LTD).

Since TC1, TC2 and TC4 activity lasts only 500 ms after the cue (S2 Fig), we consider each trial as 500 ms of simulation. We randomly generated 500 trials of Go and No-go cues. Overall,

we conducted 10 simulations with varied initial conditions, and the metrics such as firing rates and lick rates were averaged over these 10 simulations.

**Cerebellar spiking neural network model.** To efficiently simulate the cerebellar spiking neural model, we utilized the Leaky-Integrate-and-Fire (LIF) neuron model provided by CARLsim [83]. This model includes two groups of neurons within the PC-CN-IO circuitry, corresponding to the $TC_{Go}$ ($PC_{Go}$ - $CN_{Go}$ - $IO_{Go}$) and $TC_{Nogo}$ ($PC_{Nogo}$ - $CN_{Nogo}$ - $IO_{Nogo}$) groups. Additionally, sensory inputs of auditory cues were transmitted, via mossy fibers, to granule cells independently for Go ($GrC_{Go}$) and No-go ($GrC_{Nogo}$) cues. Each neuron was modeled using the LIF framework, and synaptic inputs were generated by a Poisson spike generator (see details below). The model parameters are listed in Table 1.

**Synaptic connections.** We employed a current-based mode for synaptic connections, where the total synaptic current $I_j^{syn}$ at the postsynaptic neuron $j$ due to a spike from presynaptic neuron $i$ is given at any point in time by:

$$I_j^{syn} = \sum_{i=1}^{N} s_{ij} w_{ij}$$

where $s_{ij}$ is 1 if the neuron is spiking and 0 otherwise, $w_{ij}$ is the strength of the synaptic weight between postsynaptic neuron $j$ and presynaptic neuron $i$, and N is the total amount of presynaptic connections. All the synaptic connections in the model are listed in Table 2, with a fixed delay of 1 ms for each connection.

**Sensory and error inputs.** There are two sources of synaptic inputs to the model: the sensory input (i.e., cue) and the error derived from licking behaviors. Sensory input (Go or No-go) is conveyed by 100 mossy fibers, with inter-spike intervals following a Poisson process at a rate of 4 Hz. Note that there are two sets of 100 MFs for Go ($MF_{Go}$) and No-go ($MF_{Nogo}$) cues, respectively, with each set 5% randomly connected to $GrC_{Go}$ and $GrC_{Nogo}$ (Table 2). Note that MFs also transmit the sensory input to CN.

For simulating the error in Go/No-go tasks, we assumed that the reward size is proportional to the lick rate, the model's output. Consequently, the model learns to maximize rewards by increasing the lick rate to a theoretical maximum for Go cues and to minimize penalties by decreasing the lick rate toward zero for No-go cues. The error rate resulting from

**Table 1. Parameters of LIF neuron model.**

| Group | Neuron type | # neurons | Synaptic type | $\tau_m$ | $\tau_{ref}$ | $V_{th}$ | $V_{reset}$ |
|---|---|---|---|---|---|---|---|
| $TC_{Go}$ | $GrC_{Go}$ | 2000 | Excitatory | 10 | 5 | -67 | -70 |
| | $PC_{Go}$ | 100 | Inhibitory | 10 | 5 | -67 | -70 |
| | $CN_{Go}$ | 100 | Inhibitory | 10 | 5 | -67 | -70 |
| | $IO_{Go}$ | 10x10 | Excitatory | 10 | 50 | -65 | -70 |
| | $MF_{Go}$ | 100 | Excitatory | Poisson spike generator ($\lambda$ =4 Hz) | | | |
| | $Error_{Go}$ | 100 | Excitatory | Poisson spike generator ($\lambda_{No}$) | | | |
| $TC_{Nogo}$ | $GrC_{Nogo}$ | 2000 | Excitatory | 10 | 5 | -67 | -70 |
| | $PC_{Nogo}$ | 100 | Inhibitory | 10 | 5 | -67 | -70 |
| | $CN_{Nogo}$ | 100 | Inhibitory | 10 | 5 | -67 | -70 |
| | $IO_{Nogo}$ | 10x10 | Excitatory | 10 | 50 | -65 | -70 |
| | $MF_{Nogo}$ | 100 | Excitatory | Poisson spike generator ($\lambda$ =4 Hz) | | | |
| | $Error_{Nogo}$ | 100 | Excitatory | Poisson spike generator ($\lambda_{Nogo}$) | | | |

$\tau_m$: Membrane time constant in ms; $\tau_{ref}$: absolute refractory period in ms; $V_{thr}$: Threshold voltage for firing; $V_{reset}$: Membrane potential resets to this value immediately after spike.

**Table 2. Synaptic connections.**

| Group | Presynaptic | Postsynaptic | Connection type | Initial weight |
|---|---|---|---|---|
| $TC_{Go}$ | $MF_{Go}$ | $GrC_{Go}$ | Random (5%) | 10 |
| | $MF_{Go}$ | $CN_{Go}$ | Full | 10 |
| | $GrC_{Go}$ | $PC_{Go}$ | Full | 10 |
| | $GrC_{Nogo}$ | $PC_{Go}$ | Full | 10 |
| | $PC_{Go}$ | $CN_{Go}$ | One-to-One | 50 |
| | $Error_{Go}$ | $IO_{Go}$ | One-to-One | 70 |
| | $IO_{Go}$ | $IO_{Go}$ | Gaussian | 10 |
| | $IO_{Go}$ | $PC_{Go}$ | One-to-One | 50 |
| $TC_{Nogo}$ | $MF_{Nogo}$ | $GrC_{Nogo}$ | Random (5%) | 10 |
| | $MF_{Nogo}$ | $CN_{Nogo}$ | Full | 10 |
| | $GrC_{Go}$ | $PC_{Nogo}$ | Full | 10 |
| | $GrC_{Nogo}$ | $PC_{Nogo}$ | Full | 10 |
| | $PC_{Nogo}$ | $CN_{Nogo}$ | One-to-One | 50 |
| | $Error_{Nogo}$ | $IO_{Nogo}$ | One-to-One | 70 |
| | $IO_{Nogo}$ | $IO_{Nogo}$ | Gaussian | 10 |
| | $IO_{Nogo}$ | $PC_{Nogo}$ | One-to-One | 50 |

Connection type: Full – all presynaptic neurons connect to all postsynaptic neurons; One-to-One – each presynaptic neuron connects to only one postsynaptic neuron; Gaussian – the neurons were connected with the weight proportional to their relative distance in the lattice structure. Random –presynaptic neurons randomly connect to postsynaptic neurons with a probability $p$.

licking behaviors was computed as the difference between the targeted lick rate and the actual lick rate. To achieve this, we first assumed that the lick rate was a function of the combined firing rates of $CN_{Go}$ and $CN_{Nogo}$ neurons (Fig 6B):

$$rate_{lick} = rate_{max} - \frac{rate_{max}}{1 + exp\left\{-0.3\left(CN_{Go} + CN_{Nogo}\right) + 16\right\}}$$

where $rate_{lick}$ is the lick rate, $rate_{max}$ = 6 Hz is the maximum lick rate in a trial, and $CN_{Go}$ and $CN_{Nogo}$ denote firing rate of $CN_{Go}$ and $CN_{Nogo}$ neurons, respectively. The error rate was then computed for Go and No-go cues as:

$$error_{Go} = rate_{max} - rate_{lick} \ for \ Go$$

$$error_{Nogo} = -rate_{lick} \ for \ Nogo$$

According to this equation, the error is initially positive for Go trials and negative for No-go trials, and it will converge to zero for both cues during the learning process. Finally, the error is transmitted one-by-one to the IO neurons by a Poisson spike generator group, with the rate negatively correlated with the error (Fig 6C):

$$\lambda_{Go} = -\frac{4}{3} \times error_{Go} + 6$$

$$\lambda_{Nogo} = -\frac{5}{6} \times error_{Nogo} + 2$$

To maintain the spontaneous firing level of IO neurons, $IO_{Go}$ neurons receive synaptic inputs during No-go trials, and $IO_{Nogo}$ neurons receive synaptic inputs during Go trials, keeping their firing rate around 1 Hz.

**Effective coupling between IO neurons.** The effective coupling between the IO neurons, located in a 10x10 lattice structure, was modeled using Gaussian connectivity, based on their relative distances (Fig 6D). For CN-IO inhibition, we established one-to-one connections between CN and IO neurons, with the coupling strength was modified by:

$$g_t^{cue} - g_0 = -\beta \times (CN_{t-1}^{cue} - CN_0)$$

Here, $g_t^{cue}$ is the coupling strength at trial $t$-th for a given cue (Go or No-go), $CN_{t-1}^{cue}$ denotes the firing rate of the CN at the *(t-1)*-th of the same cue condition, $g_0$ and $CN_0$ are the initial values of the effective coupling and CN firing rate, respectively, and $\beta$ is the proportional constant ($\beta = 0.8$ for $IO_{Go}$ and $\beta = 0.5$ for $IO_{Nogo}$). Note that effective coupling cannot be determined online (within the same trial) by CN activity, as the model configuration (e.g., setting the coupling strength) in the CARLsim framework must be completed before the simulation runs. Thus, the CN activity from the previous trial of the same cue condition was used.

**Bidirectional plasticity at PF-PC synapses.** We hypothesized bidirectional plasticity at PF-PC synapses of $TC_{Go}$ and $TC_{Nogo}$ neurons based on observations showing these neurons bidirectionally modulating their firing rates during learning (see S2 Fig). This bidirectional plasticity is captured by the following function:

$$\Delta w_{pf-PC} = -\alpha \left( IO - \overline{IO} \right)$$

where $\Delta w_{pf-PC}$ represents the change in weights of GrC-PC connections, *IO* denotes the firing rate of IO neurons, $\alpha$ is the learning rate ($\alpha = 0.01$ for $PC_{Go}$ and $\alpha = 0.05$ for $PC_{Nogo}$ neurons). Here $\overline{IO}$ is a constant that determines the direction of plasticity. Specifically, with the initial IO rate at approximately 3 Hz, long-term potentiation (LTP) and long-term depression (LTD) could be achieved by setting $\overline{IO} = 8$ Hz for $TC_{Go}$ and $\overline{IO} = 1$ Hz for $TC_{Nogo}$, respectively.

**Parameter tuning.** Due to a large gap between the number of model parameters and available real data in Go/No-go tasks, our objective was not precise data fitting. Instead, we manually adjusted parameters to allow the model to qualitatively reproduce lick rates and CS firing rates from our study. Initially, the lick rate was around 3 Hz, reaching a peak of 6 Hz after Go cues and dropping to 0 Hz following No-go cues (corresponding to correct-rejection trials) after learning (see S5A Fig). To replicate neural firing rates, parameters for leaky integrate-and-fire (LIF) neurons and synaptic connections were selected to ensure consistency with documented rates of Purkinje cells (PC, 40–80 Hz), cerebellar nuclei (CN, 0–30 Hz), and inferior olive (IO, 0–10 Hz) neurons [84]. Other parameters, such as weight updating constants and coefficients for lick rate and error computations, were manually adjusted to simulate licking behaviors and error signals in Go/No-go tasks.

## Statistics

All statistical analyses were performed using MATLAB software. Unless otherwise stated, data are presented as means ± standard deviation. For evaluation of correlations between neuronal response and single behavior variables (Figs 4B-F), we fitted a linear mixed-effects model with fixed effect for behavior variables and mouse index as random intercept. The analysis was conducted using MATLAB function *fitlme*, with the slope and its significance p-value were reported. Significance level: n.s, $p > 0.05$; * $p < 0.05$; ** $p < 0.01$; *** $p < 0.001$; **** $p < 0.0001$.

## Code availability

The customized MATLAB code for the analyses and the simulation using the CARLsim framework are publicly available on GitHub at the following link: https://github.com/hoang-atr/go_nogo.

## Supporting information

**S1 Fig. Regression analysis of TC activity and behavior variables in individual animals and top TC neurons.** A: sCCA of TC score and behavior variables were conducted for individual animals. Bar and lines indicate the averaged and std of CCA coefficients across 17 animals. B: we sampled the top 300 neurons for each TC at each learning stage, then removed overlapping ones (see [26] for details). As a result, we selected 2,096 neurons (termed "topTC neurons") and individually conducted PLSR of their TC score with the behavior variables. Bar and lines indicate the averaged and std of VIP score for topTC1-4 neurons. The horizontal dashed line indicates VIP score = 1. Color convention is the same as Fig 2.
(TIFF)

**S2 Fig. This figure plots PSTHs in the four cue-response conditions of topTC neurons for the four tensor components (refer to S1 Fig for neural sampling).** Blue, green and red traces are for 1st, 2nd and 3rd learning stages, respectively.
(TIFF)

**S3 Fig. Shuffled correlations between TC1 and TC2 score with reward-prediction errors δQ.** A: For each animal, we shuffled the pairs of TC1 score vs. reward-prediction errors δQ in HIT trials and conducted the regression analysis similar to Fig 4C. The bar-plots indicated frequency of the slope (left) and p-value (right) of the regression for 1000 shuffled times. B: similar to A but for TC2 score vs. δQ in FA trials. Note that, for the original data, the slope of regression was -0.65 for TC1-HIT and -1.24 for TC2-FA (vertical red lines) with p-value < 0.00001 (reported in Fig 4C&D).
(TIFF)

**S4 Fig. Firing rate in 500 trials of the neurons in the PC-CN-IO circuitry for TC$_{Go}$ (upper) and TC$_{Nogo}$ (lower) groups.**
(TIFF)

**S5 Fig. Experimental data. A: Lick rate of 17 animals during Go (light blue traces) and No-go (light red traces) trials across 7 sessions.** The thick lines represent the average lick rate across all animals. B:Summary of top TC neurons in the real data. Bars with error bars indicate the mean ± s.e.m of CS rate during Go (blue bars) and No-go (red bars) trials for topTC1 and topTC2 neurons (refer to S1B Fig for neuron sampling) in the first stage (open bars, fraction correct < 0.6) and the third stage (filled bars, fraction correct > 0.8). The differences in CS rate between the 1st and 3rd stages were significant (p < 0.0001) for all conditions. Note that both lick rate and CS rate were measured within 0-0.5 s after cue onset.
(TIFF)

**S6 Fig. Sparse CCA with TC1-4 scores computed from synchronized spikes.**
(TIFF)

**S7 Fig. Sparse CCA for TC1-4 scores with auditory cue and motor variables independent.** In these analyses, the auditory cue (Cue=1 for Go and Cue=0 for No-go cues) was made independent from motor (licking) variables, resulting in a total of 8 exploratory variables (Cue, R, Q, δQ, ELick, RLick, LLick and lick latency fluctuation). sCCA selected 5 exploratory

variables, with only a small positive correlation between TC2 and early lick count. Color convention is the same as Fig 2 with a new black column corresponding to the auditory cue.
(TIFF)

**S8 Fig. BIC score, estimated for 17 mice, of the five Q-learning models with different numbers of hyper-parameters.** The total BIC score was 21,235; 21,291; 21,414; 21,532 and 69,325 for Model #1-5, respectively.
(TIFF)

## Author contributions

**Conceptualization:** Huu Hoang, Masanori Matsuzaki, Masanobu Kano, Keisuke Toyama, Kazuo Kitamura, Mitsuo Kawato.

**Data curation:** Shinichiro Tsutsumi, Kazuo Kitamura.

**Formal analysis:** Huu Hoang, Mitsuo Kawato.

**Funding acquisition:** Masanori Matsuzaki, Masanobu Kano, Kazuo Kitamura.

**Investigation:** Huu Hoang, Shinichiro Tsutsumi, Kazuo Kitamura, Mitsuo Kawato.

**Methodology:** Huu Hoang, Mitsuo Kawato.

**Software:** Huu Hoang.

**Supervision:** Kazuo Kitamura, Mitsuo Kawato.

**Validation:** Huu Hoang, Shinichiro Tsutsumi, Mitsuo Kawato.

**Visualization:** Huu Hoang.

**Writing – original draft:** Huu Hoang, Shinichiro Tsutsumi, Masanori Matsuzaki, Masanobu Kano, Keisuke Toyama, Kazuo Kitamura, Mitsuo Kawato.

**Writing – review & editing:** Huu Hoang, Shinichiro Tsutsumi, Masanori Matsuzaki, Masanobu Kano, Keisuke Toyama, Kazuo Kitamura, Mitsuo Kawato.

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
