## [Decision Letter · Decision Letter 0]

22 Jan 2025

PCOMPBIOL-D-24-01559

Predictive reward-prediction errors of climbing fiber inputs integrate modular reinforcement learning with supervised learning

PLOS Computational Biology

Dear Dr. Hoang,

Thank you for submitting your manuscript to PLOS Computational Biology. After careful consideration, we feel that it has merit but does not fully meet PLOS Computational Biology's publication criteria as it currently stands. Therefore, we invite you to submit a revised version of the manuscript that addresses the points raised during the review process.

The reviewers provide a variety of constructive suggestions on how to improve the communication of your paper, particularly with respect to the interpretation of the results and the clear demarcation from your earlier studies.

Please submit your revised manuscript within 30 days Mar 24 2025 11:59PM. If you will need more time than this to complete your revisions, please reply to this message or contact the journal office at ploscompbiol@plos.org. Please include the following items when submitting your revised manuscript:

We look forward to receiving your revised manuscript.

Kind regards,

Abigail Morrison

Academic Editor

PLOS Computational Biology

Lyle Graham

Section Editor

PLOS Computational Biology

**Journal Requirements:**

At this stage, the following Authors/Authors require contributions: Huu Hoang. Please ensure that the full contributions of each author are acknowledged in the "Add/Edit/Remove Authors" section of our submission form.

3) Please insert an Ethics Statement at the beginning of your Methods section, under a subheading 'Ethics Statement'. It must include:

i) The full name(s) of the Institutional Review Board(s) or Ethics Committee(s)

ii) The approval number(s), or a statement that approval was granted by the named board(s).

5) We notice that your supplementary Figures are included in the manuscript file. Please remove them and upload them with the file type 'Supporting Information'. Please ensure that each Supporting Information file has a legend listed in the manuscript after the references list.

Potential Copyright Issues:

i) Figure 1A. Please confirm whether you drew the images / clip-art within the figure panels by hand. If you did not draw the images, please provide (a) a link to the source of the images or icons and their license / terms of use; or (b) written permission from the copyright holder to publish the images or icons under our CC BY 4.0 license. Alternatively, you may replace the images with open source alternatives. See these open source resources you may use to replace images / clip-art:

7) Please amend your detailed Financial Disclosure statement. This is published with the article. It must therefore be completed in full sentences and contain the exact wording you wish to be published.

8) Your current Financial Disclosure states, " HH, KK and KT were supported by Grants-in-Aid for Scientific Research in Innovative Areas (17H06313) and for Transformative Research Areas (22H05160 to M.M., 22H05156 to M Kawato, and 22H05161 to KK). HH and KT were partially supported by JST ERATO (JPMJER1801, "Brain-AI hybrid"). HH, M Kawato, and KK were partially supported by the Grant Number JP21dm0307002, JP21dm0307008, and JP19dm0207080, respectively, Japan Agency for Medical Research and Development (AMED). M Kawato was partially supported by Innovative Science and Technology Initiative for Security Grant Number JP004596, Acquisition, Technology & Logistics Agency (ATLA), Japan. M Kano and KK were partially supported by Grants-in-Aid for Scientific Research (JP18H04012, JP20H05915, JP21H04785 to M.Kano and JP22H00460 to KK) from the Japan Society for the Promotion of Science (JSPS). ".

However, your funding information on the submission form indicates receiving only one fund. Please ensure that the funders and grant numbers match between the Financial Disclosure field and the Funding Information tab in your submission form. Note that the funders must be provided in the same order in both places as well.. 

Please indicate by return email the full and correct funding information for your study and confirm the order in which funding contributions should appear. Please be sure to indicate whether the funders played any role in the study design, data collection and analysis, decision to publish, or preparation of the manuscript.

**Reviewers' comments:**

Reviewer's Responses to Questions

Reviewer #1: The manuscript describes a combined experimental and computational study of reinforcement learning in the cerebellum. The research integrates a Q-learning model with experimental spike detection, Tensor Component Analysis (TCA) of the neuronal activity, correlation analyses of the neuronal activity against reinforcement learning and sensorimotor control variables, and a spiking neural network model of the olivo-cerebellar system. This is excellent work and beautifully presented – I recommend publication in PLOS Computational Biology subject to minor revisions.

Minor comments:

1. In the neural network model, the lick rate is modelled as a sigmoid function of the combined spike rate of cerebellar nucleus (CN) neurons, with decreasing lick rates for increasing CN spike rates (and a maximum lick rate at a CN spike rate of zero). This differs from many other cerebellar models (e.g. Medina et al. J Neurosci. 2001) that assume movements are triggered by increased output from the CN. It would be good to discuss the neural circuitry downstream of the CN that could implement this mapping of decreased (including zero) CN output on increased lick rates.

2. Although this is already briefly mentioned in the discussion, the network model doesn’t represent a few important anatomical features, in particular, the convergence of many Purkinje cells on CN neurons. It would be good to discuss the expected effect of these features in more detail.

3. The neural network only represents TC1 and TC2, but not TC3 and TC4, although they are also (positively / negatively) correlated with the reward and licking. Would it be possible to speculate about a neural network (without additional simulations) that includes TC3 and TC4?

4. Line 155: “CSs in HIT trials (n = 3,788) were large and distributed across the entire medial hemisphere at the initial learning stage, but later on (2nd and 3rd stages), they became stronger and compartmentally focused on positive zones.” – Figure 2 A shows that positive zones (in particular, 5+ and 5a+) already stand out in the initial learning stage.

5. Figure 2 A also shows that the CS activity increases during learning (from stage 1 to stage 3) in CR trials?

6. Line 180: “We found that neurons in the lateral hemisphere were associated with Q and dQ, but not the reward R per se (Fig 2B).” – Figure 2 B indicates that the lateral zone 7+ is also strongly associated with R.

7. Line 334: “two groups of neurons, corresponding to TC1 and TC2, operating within a modular architecture where each group independently processes Go and No-go cues” - strictly speaking, only the climbing fibre input represents Go and No-go cues independently, the mossy fibre input to the TC1 and TC2 groups seems to be shared.

8. Please check the grammar of a few sentences, e.g. line 119 “As temporal evolution …”, line 576 “while the probability …”.

9. Line 394: “IO firings”  IO spiking or IO firing.

Reviewer #2: This is a nice study from Dr Hoang and colleagues examining the information content of complex spike Ca responses across large populations of Purkinje neurons during a go-no-go task. The data have been previously published, but the study here extends the analyses employed in those previous studies to examine the whether the observed patterns of CS activity can be used in a Q-learning reinforcement learning algorithm. Further, they advance a straightforward computational model that accounts for the distinct modules they observe with CS tuning. I think the study represents a nice advance. I do not have any substantive scientific suggestions. However, I think it is important for the authors to explicitly spell out in the main text what is new and what has previously been published. Some of the figure panels are identical (or practically identical, with different color lookup tables applied) to previously published work for the same group (e.g. Hoang et al., 2023). In reading that manuscript there is significant overlap with the present study, but nevertheless the present study is distinct and worthy of separate publication. That said, a more explicit enumeration of the distinctions between the two studies will be a service to the readership.

Minor

1. In lines 156-159 I would suggest that the authors call the responses CSs instead of PSTHs.

2. In Hoang et al., 2023 the authors describe ‘error’ responsive CSs and reward-responsive CSs. I found this language helpful to classify different types of responses. Here the description of CS tuning does not use these descriptions, even for CSs that are negatively correlated with reward, or responsive to false alarms. It would be helpful to better match the descriptions in the two papers or to provide an explanation as to why the previous nomenclature no longer holds.

**Have the authors made all data and (if applicable) computational code underlying the findings in their manuscript fully available?**

Reviewer #1: Yes

Reviewer #2: None

PLOS authors have the option to publish the peer review history of their article (what does this mean? ). If published, this will include your full peer review and any attached files.

**Do you want your identity to be public for this peer review?** For information about this choice, including consent withdrawal, please see our Privacy Policy .

Reviewer #1: **Yes: ** Volker Steuber

Reviewer #2: No

**Figure resubmission:**
---

## [Decision Letter · Decision Letter 1]

21 Feb 2025

Dear Dr. Hoang,

We are pleased to inform you that your manuscript 'Predictive reward-prediction errors of climbing fiber inputs integrate modular reinforcement learning with supervised learning' has been provisionally accepted for publication in PLOS Computational Biology.

Best regards,

Abigail Morrison

Academic Editor

PLOS Computational Biology

Lyle Graham

Section Editor

PLOS Computational Biology

Reviewer's Responses to Questions

**Comments to the Authors:**

Reviewer #1: The authors have addressed all of my comments.

**Have the authors made all data and (if applicable) computational code underlying the findings in their manuscript fully available?**

Reviewer #1: Yes

PLOS authors have the option to publish the peer review history of their article (what does this mean? ). If published, this will include your full peer review and any attached files.

**Do you want your identity to be public for this peer review?** For information about this choice, including consent withdrawal, please see our Privacy Policy .

Reviewer #1: **Yes: ** Volker Steuber

---

## [Editor Report · Acceptance letter]

PCOMPBIOL-D-24-01559R1

Predictive reward-prediction errors of climbing fiber inputs integrate modular reinforcement learning with supervised learning

Dear Dr Hoang,

I am pleased to inform you that your manuscript has been formally accepted for publication in PLOS Computational Biology. Your manuscript is now with our production department and you will be notified of the publication date in due course.

With kind regards,

Zsofia Freund
